# Delineation of Dew Formation Zones in Iran Using Long-Term Model Simulations and Cluster Analysis

Nahid Atashi[1,2], Dariush Rahimi[1], Victoria A. Sinclair[2], Martha A. Zaidan[2,3], Anton Rusanen[2], Henri Vuollekoski[2], Markku Kulmala[2,3,4,5], Timo Vesala [2,6,7], Tareq Hussein [2,8]

[1] Faculty of Geographical science and Planning, University of Isfahan, Isfahan, 8174673441, Iran

[2] Institute for Atmospheric and Earth System Research (INAR/Physics), Faculty of Science, University of Helsinki, Helsinki, FI-00014, Finland.

[3] Joint International Research Laboratory of Atmospheric and Earth System Sciences, School of Atmospheric Sciences, Nanjing University, Nanjing 210023, China.

[4] Aerosol and Haze Laboratory, Beijing Advanced Innovation Center for Soft Matter Science and Engineering, Beijing University of Chemical Technology, Beijing 100029, China.

[5] Faculty of Geography, Lomonosov Moscow State University, 119991 Moscow, Russia.

[6] Institute for Atmospheric and Earth System Research (INAR/Forest), Faculty of Agriculture and Forestry, University of Helsinki, Helsinki, FI-00014, Finland.

[7] Research Education Center of Environmental Dynamics and Climate Change, Yugra State University, 628012 Khanty-Mansiysk, Russia

[8] School of Science, Department of Physics, University of Jordan, Amman, 11942, Jordan.

*Correspondence to*: Tareq Hussein (tareq.hussein@helsinki.fi) and Timo Vesala (timo.vesala@helsinki.fi)

**Abstract.** Dew is a non-conventional source of water that has been gaining interest over the last two decades, especially in arid and semi-arid regions. In this study, we performed a long-term (1979–2018) energy balance model simulation to estimate dew formation potential in Iran aiming to identify dew formation zones and to investigate the impacts of long-term variation in meteorological parameters on dew formation. The annual average of dew occurrence in Iran was ~ 102 days, with the lowest number of dewy days in summer (~ 7 days) and highest in winter (~ 45 days). The average daily dew yield was in the range of $0.03 – 0.14$ L/m$^2$ and the maximum was in the range of $0.29 – 0.52$ L/m$^2$. Six dew formation zones were identified based on cluster analysis of the time series of the simulated dew yield. The distribution of dew formation zones in Iran was closely aligned with topography and sources of moisture. Therefore, the coastal zones in the north and south of Iran (i.e., Caspian Sea and Oman Sea), showed the highest dew formation potential with 53 and 34 L/m$^2$/year, whereas the dry interior regions (i.e., central Iran and the Lut Desert), with the average of 12-18 L/m$^2$/year had the lowest potential for dew formation. Dew yield estimation is very sensitive to the choice of the heat transfer coefficient . The uncertainty analysis of the heat transfer coefficient using eight different parameterization revealed that the parameterization used in this study – the Richards (2009) formulation - gives estimates that are similar to the average of all methods and are neither much lower nor much higher than the majority of other parameterizations and the largest differences occur for the very low values of daily dew yield. Trend analysis results revealed a significant ($p < 0.05$) negative trend in the yearly dew yield in most parts of Iran during the last 4 decades (1979-2018). Such a negative trend in dew formation is likely due to an increase in air temperature and a decrease in relative humidity and cloudiness over 40 years.

## 1 Introduction

Scarcity and continuously increasing demand on freshwater is one of the socio-economic problems in many countries, especially in arid and semi-arid regions. It is anticipated that two-thirds of the world's population will suffer of freshwater shortage by the year 2025 (Human Development Report, 2006). In fact, the water crisis will not only be limited to freshwater resources but also will have an extreme impact on agriculture and livestock (Madani, 2005).

Scientists have also warned that water shortage will continue further in the coming decades in the Middle East, where water is one of the most valuable and vulnerable natural resources (Mehryar et al. 2015; Ashraf et al. 2019., Bozorg-Haddad et al. 2020). Iran is one of these countries suffering of freshwater shortage and climate change consequences (Karimi et al., 2018., Ashraf and Fahimi, 2019; Emami and Koch, 2019; Naderi, 2020). For instance, the annual average rainfall in Iran is about 250 mm (Alizadeh, 2011). Besides that, 65% of the country is arid, 20% is semi-arid, and only 15 % has a humid and semi-humid climate. The Iranian Annual Renewable Water Resources is currently less than 2000 $m^3$/capita and with the current population growth rate (~1.19 %; CIA, 2018), is expected to be reduced to be less than 1000 $m^3$/capita in 2025 (Madani, 2005; Moradi, 2017). Therefore, looking for alternative resources of freshwater is a necessity in the arid and semi-arid regions in Iran.

The atmosphere can be considered a huge renewable reservoir of water (i.e., cloud, fog, and water vapor) and enough to meet the needs of every person on the planet (Tu et al., 2018). Dew is a non-conventional atmospheric resource of water, which forms during phase transition from vapor to liquid (Tomaszkiewicz et al., 2015), or condensation of atmospheric water vapor on surfaces with temperature below dew point (Khalil et al., 2016). Although the amount of dew that can be harvested is relatively small, it can enhance water supply in certain climates/regions, particularly in the absence of precipitation (Tomaszkiewicz et al., 2015). Extracting dew water as a sustainable natural phenomenon by means of radiative (or passive) condensers has been gaining interest over the last two decades. Research on radiative condensers started in the early 1960s with a study conducted in Negev Desert by Gindel (1965). Based on studies in different locations worldwide (Table 1), the highest amount of daily dew yield (typically in the range of 0.2- 0.6 $L/m^2$ was observed in arid deserts and semi-arid areas (Kidron, 1999; Alnaser, 2000; Agam and Berliner, 2006; Sharan et al., 2007; Lekouch et al., 2012; Tomaszkiewicz et al., 2017; Jia et al., 2019; Tuurre et al., 2019). Some regions with humid climates (e.g., coastal areas and Islands) showed lower yield (~ 0.2- 0.4 $L/m^2$) (Sharan, 2005; Clus et al., 2008; Museli et al., 2002 and 2009; Hanisch et al., 2015), and urban environments had the minimum dew yield (~ 0.02- 0.3 $L/m^2$) (Richards, 2004; Beysens et al., 2006; Ye et al., 2007; Muskała et al., 2015; Odeh et al., 2017).

Despite the importance of dew and its potential especially in dry areas, it has been disregarded from the water budget in Iran (e.g., Esfandiarnejad et al., 2010; Davtalab et al., 2013). There is a lack of dew data in Iran; therefore, we utilized a gridded model (Vuollekoski et al., 2015) and performed simulations covering 40 years (1979–2018) to estimate the potential of dew yield. This model is based on an energy balance similar to models used in previous studies (e.g., Nilsson, 1996; Jacobs, 2009; Maestre-Valero et al., 2011; Arias-Torres and Flores-Prieto. 2016; Beysens, 2016) conducted in different environments. Previous studies have demonstrated that energy balance models are able to predict dew yield within a reasonable agreement

with measured dew yield and could be also applicable elsewhere. For example, Tomaszkiewicz et al. (2016), applied a dew prediction model that was developed by Beysens (2016), to generate a dew yield atlas for the Mediterranean region (142 stations). The objective of this study is to identify the major dew formation zones in Iran using a long-term model simulation and to investigate the possible impacts of historic changes to the climate over the last 40 years on dew in Iran.

## 2.Methods

In order to estimate dew collection potential in Iran, we combined a computationally efficient dew formation model with meteorological reanalysis data spanning 40 years. The model simulation results were used to investigate the spatial-temporal variation of dew yield in Iran. In this study, the term "dew yield " refers to the amount of water that can be harvested on a $1m^2$ condenser.

### 2.1 Meteorological input data

The dew formation model (which is described in detail in section 2.2) requires meteorological data as input. In Iran, there are very few stations with long-term observations of all the required meteorological variables. Therefore, instead of driving the dew model with observations, we use ERA-Interim (Berrisford et al., 2011; Dee et al., 2011), which is a meteorological global reanalysis produced by the European Centre for Medium-Range Weather Forecasts (ECMWF). Reanalysis combines a massive number of observations from a number of sources (satellite, radiosondes, aircraft, buoy data, stations, etc.) with a numerical weather prediction model to produce a coherent, long-term gridded data set of the atmospheric dynamic and thermodynamic state over the whole globe (Tompkins, 2017).

ERA-Interim covers the time period from January 1979 until August 2019, has a native resolution of 0.75 degrees, which is approximately 80 km, and 60 model levels in the vertical profile. Here we considered the time period during 1979–2018 and used input data interpolated to a grid resolution of 0.25 degrees (~30km) over a domain covering all parts of Iran (Figure 1). This interpolation was done during the download process using standard ECMWF procedures: continuous fields (e.g. temperature, precipitation) were interpolated using bilinear interpolation and discrete fields (e.g. vegetation, soil type) were interpolated using a nearest neighbor approach.

Similar to all atmospheric reanalysis, ERA-Interim contains two distinct types of fields: analysis fields and forecast fields. The analysis fields were produced by combining a very short-range forecast and observations to produce the best fit for both. The forecast fields were produced by the numerical forecast model starting from an analysis. In ERA-Interim, the analysis fields were available every 6 hours (00:00, 06:00, 12:00, and 18:00 UTC) and the forecast fields were available every 3 hours and hence can be used to fill in the gaps between the analysis. Furthermore, the forecast fields can be either instantaneous or accumulated over the forecast period.

The variables that are required for the dew formation model are: air temperature ($T_a$), dew point temperature ($D_P$), wind speed (*WS*), short-wave ($R_{sw}$), and long-wave solar radiation ($R_{lw}$). From ERA-Interim we extracted the 2-m $T_a$ and $D_P$ from both the analysis and the instantaneous forecasts and obtain the short-wave and long-wave surface radiation as accumulated forecast fields. To obtain the mean value over each time interval, the difference of the accumulated values between two consecutive time steps was taken and then divided by the time difference in seconds. The wind speed at 2m was not directly available from ERA-Interim; therefore, we obtained the wind components (*U* and *V*) at 10 m and the surface roughness ($z_0$ – an instantaneous forecast field) and assumed a logarithmic winds profile to obtain its values at 2-m according to

$$WS = \frac{log\left(\frac{2+z_0}{z_0}\right)}{log\left(\frac{10+z_0}{z_0}\right)} \sqrt{U_{10}^2 + V_{10}^2}, \tag{1}$$

where $z_0$ is the surface roughness and $U_{10}$ and $V_{10}$ are the horizontal wind speed components at 10 meters. It is important to understand that the logarithmic assumption is only strictly valid during neutral stability conditions. During stable conditions (such as during night time) it overestimates the 2-m wind speed whereas in unstable conditions it underestimates the 2-m wind speed (Riou, 1984; Holtslag, 1984; Petersen et al., 1998; Oke, 2002; Optis et al., 2016).

**2.2 Dew formation model description and output**

The global dew formation model used in this study was originally developed by Vuollekoski et al. (2015) to estimate dew potential. The approach is similar to Pedro and Gillespie (1982) and Nikolayev et al. (1996). The model reads all input data (described in section 2.1) for a given grid point and numerically solves the mass and heat balance equation by using a fourth-order Runge-Kutta algorithm with a 10s time step (i.e., The ERA-Interim data from 3 hourly resolution were linearly interpolated to obtain 10-second resolution). The mass and heat energy balance model is written as:

$$\frac{dT_c}{dt}(C_c m_c + C_w m_w + C_i m_i) = P_{rad} + P_{cond} + P_{conv} + P_{lat} \tag{2}$$

where $dT_c/dt$ is the rate of change of the condenser temperature. $C_c$, $C_w$, and $C_i$ are the specific heat capacity of the condenser, water, and ice; respectively. Here, mc, mw, and mi are the mass of the condenser, water, and ice; respectively. The right-hand side of Eq. (2) describes the heat exchange involved in the process: $P_{rad}$ is the net radiation, $P_{cond}$ is the conductive heat exchange between the condenser surface and the ground, $P_{conv}$ is the convective heat exchange, and $P_{lat}$ is the latent heat released by the condensation or desublimation of water.

The model was setup so that it assumes similar conditions for the phase-change of pre-existing water or ice on the condenser sheet. For instance, if the water on the condenser is in the liquid phase (i.e., $m_w > 0$) and the condenser temperature $T_c < 0$ °C, then the sheet is losing energy (i.e., the right-hand side of equation (2) is negative). In that case, instead of solving Eq. (2), Tc is assumed to be constant and the lost mass from the liquid phase of water is transferred to the cumulated mass of ice; i.e., the water is transformed from liquid phase to solid phase. Consequently, equation (2) is replaced by

$$L_{wi} \frac{dm_w}{dt} = P_{rad} + P_{conv} + P_{lat}, \tag{3}$$

where $L_{wi}$ is the latent heat of fusion. If the water on the condenser is in the solid phase (i.e., $m_i > 0$) and the condenser temperature $T_c > 0$ °C, a similar equation is assumed for the change rate of ice mass ($m_i$).

Note that Eq. (3) is not related to the condensation of water; it only describes the phase change of the already condensed water or ice on the condenser. For the water condensation rate, which is assumed independent of Eq. (3), the mass-balance equation is then assumed as

$$\frac{dm}{dt} = \max\left[\ 0\ ,\ S_c k\left(P_{sat}(T_d) - P_c(T_c)\right)\right] \tag{4}$$

where m represents either the mass of ice ($m_i$) or water ($m_w$) depending on whether $T_c$ is below or above 0 °C. $P_{sat}(T_d)$ is the saturation pressure at the dew point temperature and $P_c(T_c)$ is the vapor pressure over the condenser sheet. Here, $S_c$ is the condenser surface area and $k$ is the mass transfer coefficient,

$$k = h / L_{vw}\gamma = 0.622h / C_a p \tag{5}$$

where $L_{vw}$ is the specific latent heat of water vaporization, $\gamma$ is the psychrometric constant, $C_a$ is the specific heat capacity of air, and $p$ is the atmospheric air pressure. Here, h is the heat transfer coefficient,

$$h = 5.9 + 4.1\ u\ (511 + 294) / (511 + T_a) \tag{6}$$

where $u$ and $T_a$ are the prevailing horizontal wind speed and the ambient temperature 2 meters above the ground. This parameterization of the heat transfer coefficient is taken from Richards, (2009). However, the dew model is designed in such a manner that any functional form can be used for the heat transfer coefficient thus allowing the sensitivity of the modelled dew amounts to the formulation of the heat transfer coefficient to be assessed (see section 3.3 for such an analysis).

In practice, the wettability of the surface affects the vapor pressure $P_c$ directly above it. In other words, $P_c$ is lower over a wet surface; and thus, condensation may take place even if $T_c > T_d$. It is also assumed (in equation 4) that there is no evaporation or sublimation during daytime even if $T_c > Ta$. Furthermore, the model simulation resets the cumulative values for water and ice condensation at noon (local time) and takes the preceding maximum value of $m_w + m_i$ as the representative daily yield given in millimeter on a 1 $m^2$ condenser sheet (i.e., mm/ $m^2$/day equals to L/ $m^2$/day).

This way, the model simulation replicates the daily manual dew water collection of the condensed water around sunrise; i.e., after which $T_c$ is often above the dew point temperature. All terms and nomenclature are described in more detail in Table 2 and Table 3.

It should be noted here that, similar to many numerical models, this model has some limitations that should be considered when interpreting the results. For instance, both heat and mass coefficients are semi-empirical parameters that depend on wind speed (i.e., here we used the parameterization by Richards (2009), valid for $u < 5$ m/s). In addition, the 10s-time step in the model does not allow condensed water droplets to be eliminated on the condenser surface by evaporation. Moreover, the model predicts any dew condensation, regardless if it is collectible or not; therefore, it is expected to overestimate dew yield. The spatial data resolution is $\sim 30$ km, which limits the model's ability to resolve local microclimates, particularly in areas with complex topography where the topography can modify the large-scale winds and lead to large variations in local temperatures. However, when considering cumulative dew yield over long time periods the model performs well. Therefore, as the model

uses the meteorological gridded dataset (ERA-Interim), which is readily available for the whole globe, it can be applied anywhere in the world including other arid and semi-arid areas even if they lack observations.

## 2.3 Cluster analysis

Cluster Analysis (CA) is an effective statistical tool and technique that groups similar data points such that the points in the same group are more similar to each other than the points in the other groups. The group of similar data points is called a Cluster which can be used for various applications (Corporal-Lodangco and Leslie, 2016; Gungor and Ozmen, 2017). There are two main clustering methods: hierarchical and non-hierarchical cluster analysis. Hierarchical clustering (used in this study) combines cases into homogeneous clusters where objects at one level are combined with objects at another level and produce

clusters that are not allowed to overlap (Bunkers and Miller, 1996; Yim and Ramdeen, 2015). Two different strategies for hierarchical clustering exist: Agglomerative and divisive (Lior and Maimon, 2005). In this study, we used hierarchical agglomerative clustering (HAC, Nielsen, 2016) which starts with N clusters (i.e., here is the total number of grid points), each containing one object, and join those two objects that are most "similar". This process continues until only one cluster, containing all the data, remains (Bunkers and Miller, 1996). In order to decide which clusters should be combined (for

agglomerative), a measure of dissimilarity between sets of observations is required. The similarity measurement is a critical step in hierarchical clustering as it can influence the shape of the clusters (Nielsen, 2016). With metric data, the most commonly used distance measure (a measure of the distance between pairs of observations) is "Euclidean distance". The Euclidean distance ($d_{ij}$) between two objects $i$ and $j$ in a two-dimensional data matrix is simply the squared difference between two observations for each of $p$ variables, summed over the variables and $k$ is the number of observations (Fovell and Fovell, 1993;

Dokmanic et al., 2015). This can be written as:

$$d_{ij} = \sqrt{\sum_{k=1}^{p}(x_{ik} - x_{jk})^2} \qquad (7)$$

Here we applied this method to a two-dimensional matrix (2496×14610), where the number of rows represented the number of spatial grid points in the model simulation domain and the number of columns represented the time (i.e., cumulative daily dew yield).

After all distances were calculated, the next step is to merge the two closest entries to form a new cluster based on a linkage criterion. The linkage criterion determines the distance between sets of observations (here is the spatial grid points) as a function of the pairwise distances between observations. There are some commonly used linkage criteria: single linkage, complete linkage, average distance, and Ward's minimum variance methods, which differ in a way how the distances between entries are calculated and how the two closest entries are defined (Stooksbury & Michaels, 1991; Murtagh and Legendre,

2014). In this study, Ward's minimum variance method (Ward, 1963) is used. This method is the most frequently clustering technique used in climate research (Yokoi et al., 2011; Mimmack et al., 2000; Siraj-Ud-Doulah and Islam, 2019) and gives the most consistent clusters (Kalkstein et al. 1987). It calculates the means of all variables (the amount of dew) within each cluster,

then calculates the Euclidean distance to the cluster mean of each case, and finally sums across all grid points (Unal et al., 2003).

In any CA, the optimal number of clusters is an important issue. There is no reliable and universally accepted method to determine the optimal number of clusters. Kaufmann and Weber (1996) (see also Unal et al., 2003 and Burlando, 2009) suggested showing the total variance of subsequent merged clusters as a function of the number of remaining clusters. This information can be used as an indicator to decide the number of clusters but still, a visual check of the result can help to make the right decision. The suitable number of clusters has to be chosen somewhere in the transition between the distance values when a sudden decrease is observed as illustrated in Figure 2a. In our case, few steps at N=3, 4, 6, 7, and 10 are recommended as optimal numbers of clusters. By visualizing all these steps, N=6 was found to be the best number of clusters for this study because fewer clusters (i.e. 3 and 4 clusters) were not able to capture the different climate and dew zones. Furthermore, choosing more clusters (i.e. 7 and 10 clusters) gives some groups that replicate each other. The results of hierarchical clustering are usually presented in a dendrogram (Nielsen, 2016). The dendrogram of our 6 clusters has shown in Figure 2b.

## 3. Results

### 3.1 Spatial-temporal variation of dew occurrence and yield

According to the model simulation results (cumulative daily dew yield in the form of dew and hoarfrost), dew formation occurred almost everywhere in Iran as illustrated in Figure 3, which shows the seasonal occurrence of dew as a fraction of days with any dew yield. The frequency of dew occurrence was more than 80% (~75 days) in most areas of Iran in wintertime (December–February, Fig. 3a). The mean occurrence of dew was rather similar during spring (March–May, ~ 50 days, Figure 4b) and autumn (September–November, ~40 days, Fig. 3d) with the highest number of dew days (more than 90% (~80 days)) in the mountainous and coastal areas and the lowest (less than 40% (~35 days)) mostly in the dry interior and eastern areas. The lowest frequency of dew occurrence (i.e., less than 10 days) was in summer (June–August, Fig. 3c) when dew formation was limited to a narrow part along the Caspian Sea and the northern domains of Alborz mountain.

Limiting the dew occurrence analysis to days with dew yield $> 0.1$ L/ $m^2$ day also confirmed the seasonal characteristics of the temporal-spatial occurrence of dew. However, in this case, the frequency of dew occurrence days was less (in the range of 6–45 days for summer and winter, respectively (Fig. S1), and the spatial scale of dew formation shrank to include only a few parts of the coastal and high mountain regions during spring, summer, and autumn. This notable difference between the two maps (i.e., Fig. 3 and Fig. S1) is associated with the model setup. The model tends to forecast any dew event, regardless if it can be collectible or not. In practice, very small dew quantities are generally not harvestable as droplets remain pinned to the condenser surface and gravity cannot lead them to the collection tank.

We subsequently calculated the seasonal daily means of the cumulative dew yield (Fig. 4) which show a clear seasonal cycle with high dew yields during the winter and low yields during the summer in most parts of Iran. The monthly means of the cumulative dew yield are shown in Figure S2. Both seasonal and monthly maps show that the mountain regions had dew

occurrence throughout the year with mean cumulative daily dew in the range 0.11–0.18 L/m$^2$/day. In winter, dew occurred almost everywhere in Iran with the highest yields in the southern part of the Persian Gulf and Oman Sea coastline (mean cumulative daily dew in the range 0.15–0.23 L/m$^2$/day). In spring (i.e., April, May), a spatial pattern was observed which indicated the formation of dew was mainly parallel to the mountain range (Alborz (East- West) and Zagros (north-west and south-east)). The reason could be related to the temperature, which increases, and relative humidity which decreases during

these spring months. Therefore, in spring, in most areas conditions for dew formation were not present except in high elevation areas where the condition still favor dew formation. During summer and until the middle of autumn (i.e., July– Oct) a unique spatial pattern was evident which shows the distribution of dew formation was only limited to a narrow belt in coastal areas in the north along the Caspian Sea. In all other areas, the monthly amount of dew yield was almost zero.

## 3.2 Cluster analyses – Dew formation zones

### 3.2.1 Dew zones – a general overview

According to our Cluster Analysis (CA) summarized in Section 2.2, we identified 6 dew formation zones in Iran (Fig. 5). The amount of daily dew yield in Iran and related climatological parameters (e.g., temperature, relative humidity) for dew formation as well as the percentiles (i.e. 25%, median, 75% and 99%) of daily dew yields as averages for each cluster are listed in Table 4. As will be shown in this section, the dew formation zones in Iran are clearly aligned with topography, sources of moisture,

and climate zones. Furthermore, the mountains and seas played major roles in the spatial distribution of dew formation zones. Note that the maximum daily dew yield in this section is presented as the 99th percentile of daily dew. In order to have an insight into the climatological condition in each dew zone, we selected one synoptic station in each dew zone and investigated some related meteorological parameters (e.g., temperature, humidity, wind speed, …) in nighttime hourse (i.e. 18:00, 21:00, 00:00, 03:00), when dew formation occurs, for the time period 1980-2010 (30 years) which is shown in Figure 7 and Figure 8.

*Dew zone A – Caspian Sea region*

We identified the first dew formation zone as the "Caspian Sea region", which covered the southern shores of the Caspian Sea and the northern domains of the Alborz mountain range This dew zone includes about 7% of the total land area of Iran (Fig. 5), which also includes the largest forest area in Iran. The overall mean daily dew yield in this region was ~0.14 L/m$^2$ which was the highest among all of the dew zones and the maximum dew yield was 0.30 L/m$^2$/day (Table 4). Interestingly, this dew

zone is different compared to the other dew zones concerning the annual cycle of dew formation; in this dew zone, dew formation occurred throughout the year whereas all other zones exhibit a strong annual cycle (Fig. 6). The mean frequency of dew occurrence in this zone was more than 330 days/year. Even in summer, when dew almost vanished in other dew zones, this zone had a significant amount of dew yield (Fig. 4). The mean yearly dew yield in this region is estimated at about 53 L/m$^2$ and the maximum yield is more than 100 L/m$^2$. The high potential of dew formation in this zone during the year is due

to very suitable climatological and geographical conditions. The synoptic station Ramsar, located in this dew zone, shows the climate of dew zone A to be characterised by, low temperature, high humidity, and the smallest dewpoint depression (i.e., the

smallest difference between the temperature and dew point) along with little variation in the relative humidity and dewpoint depression throughout the year (Fig 7a). Moreover, due to being a forest area, the wind speed is relatively low (Fig, 8a), which favors dew condensation


### *Dew zone B – Zagros Mounain region*

Dew zone B included the Zagros mountain region (i.e., northern and central parts) and the eastern part of the Alborz mountains. This dew zone covered about 15% of Iran (Fig. 5) and represented a mountain climate with very cold and dry weather in winter and mild weather in summer (Fig. 7b; Zanjan station). Furthermore, due to the high elevation, the diurnal variation of

temperature within this dew zone is large. These areas receive high levels of solar radiation during the daytime and reflect it back quickly to space in the form of long-wave radiation during night-time. Therefore, the temperature drops rapidly during night-time. Enough moisture in the atmosphere, in addition to this strong nocturnal cooling, favored dew formation. The overall mean daily dew yield and variation in this region was $0.08\pm0.05$ L/m$^2$/day and the highest dew yield was 0.23 L on a 1m$^2$ condenser sheet. The highest amount of dew yield in this dew zone was observed during spring when typically, the prevailing

winds in this region are westerlies and which are accompanied by moderate to high relative humidity (Fig. 7b) and low wind speed(Fig 8b). The amount of dew yield decreased rapidly after May and was almost absent during summertime (Fig. 6). This is a result of higher temperature (i.e., due to atmosphere transparency and receiving high solar radiation) and lower relative humidity (Fig. 7b) and also the lack of efficient moisture sources in this dew zone In general, the mean frequency of dew occurrence in this zone was 63% (~245 days). The mean yearly dew yield in this zone was about 30 L/m$^2$ and the maximum

was more than 70 L/m$^2$ (Table 4).

### *Dew zone C – Central Iran*

The third dew zone is the Central Iran region. Central Iran consists of the southern slopes of the Alborz Mountains in the north, the Zagros Mountains in the south and the central Iranian ranges. These areas are mostly hot and very dry. Alborz and Zagros

mountains prevent moisture penetration from the Caspian Sea and westerlies so that the amount of water vapor pressure is very low (~7 hPa, Masudian, 2011). This zone covered about 20% of Iran and included the Kavir desert basin, Salt Lake, and some parts in the north-east (Fig. 5). The overall mean daily dew yield in this region is estimated to be about 0.05 L/m$^2$ and the maximum yield (99$^{th}$ percentile) was about 0.21 L/m$^2$/day. The average yearly dew yield in this region was about 18 L/m$^2$ and the maximum yield was less than 50 L/m$^2$/year. The dew period in this zone starts in autumn and continues until mid-

spring (i.e., October–April), however, the frequency of dew occurrence (> 0.1 L/m$^2$/day) is about 80 days. Isfahan station is located in this dew zone and is representative of the climate of this dew zone. The dew point temperatures are very low (mainly around or less than zero) all year round and have little annual cycle (Fig 7c). The relative humidity is low in spring but increases in Autumn (Fig. 7c) when temperatures start to decreases and dew formation also starts. Furthermore, the wind speed is quite weak (less than 5 m/sec, 2.5 m/sec in average, Fig. 8c) and does not have a pronounced annual cycle. Therefore, most likely

humidity and temperature are the key factor in formation of dew in this station/zone. More specifically, once relative humidity start increasing and temperature decreases (in autumn and winter), dew can also form.

*Dew zone D – Lut desert*

We identified the fourth dew zone (i.e., Dew zone D) that included the Lut desert (175,000 km$^2$; Alizadeh et al. 2014), which is an arid and hyper-arid desert (Fig. 5). This zone, with 35% of all grid points in the land areas of Iran, is the largest dew zone; however, it has the least dew occurrence (~15 days per year with dew yield > 0.1 L/m$^2$/day) and a mean yield of 0.03 L/m$^2$/day. Indeed, this part of the country includes the driest areas (i.e., water vapor pressure is < 5 hPa. Based on a survey conducted by scientists at NASA's Earth Observatory during the summer of 2003–2009 (see: Temperature of Earth:

https://www.universetoday.com/14367/planet-earth), the Lut Desert was the hottest (~71 °C) land surface on Earth, see also Khandan et al. (2018). The synoptic station Tabas is located in this dew zone and has a climate characterized by high temperatures (higher than the synoptic stations we considered in dew zones A, B and C) and low relative humidity in summer (Fig. 7d) In addition to the dryness, these areas have high diurnal variations in temperature, mostly clear sky, extremely sparse vegetation, and frequent high wind speed. In wintertime, the temperature decreases and the moisture increases (Fig. 7d), as a

result of the westerly prevailing wind and thus, this dew zone experienced its highest amount of dew yield in winter. In contrast, in the warm season (i.e., May–September) dew was almost completely absent (Fig. 6). The reason is due to high temperature, longer day time duration, and a strong north-south pressure gradient between the thermal low-pressure system over the desert lands and a cold high-pressure over the Hindu Kush mountains in northern Afghanistan (Alizadeh et al., 2014) that generates the strong summer wind called "the Sistan wind of 120 days". It was called so since it occurs during late May through late

September (about 4 months) in the east and southeast of the Iranian's Plateau, particularly the Sistan Basin. The typical wind speed of the Sistan is 30–40 km/h, but it could occasionally exceed 100–110 km/h, which impedes dew formation during the summer season. Thus, the key factors for dew condensation (high humidity, low wind speeds) are not present for most of the year in this dew zone. Consequently, the average yearly dew yield in this zone was low - about 12 L/m$^2$ and the maximum yield was about 40 L/m$^2$/year.


*Dew zone E – Persian Gulf region*

    The Persian Gulf dew zone included the coastal line of the Persian Gulf and some parts of the western half of the land areas in Iran (~9% of all grid points; Fig. 5). The overall mean daily dew yield is about 0.06 L/m$^2$, which is lower than the other coastal zones in the north (i.e., Caspian Sea; dew zone A) and south of Iran (i.e., Oman Sea; dew zone F). However, the maximum

daily dew yield in winter (i.e., December–February) was higher than that in the Caspian Sea zone. Indeed, this dew zone benefits from two huge sources of moisture (i.e., Persian Gulf and Karon river), although high temperatures (e.g. as observed at the synoptic station Ahvaz, Fig 7e) , thermal high pressure, and dry winds, especially during the warm season (i.e. May–September), do not favor the formation of dew. Therefore, , the period of dew formation was about 7 months starting in October

and ending in April. However, the frequency of dew occurrence > 0.1 L/m$^2$/day is about 117 days during November-February

(Fig. 6), when relative humidity is at its highest level and temperature and wind speed are relatively low compare to the rest of the year (Fig. 7e and Fig. 8e). The average yearly dew yield in this zone was about 24 L/m$^2$ and the maximum was > 70 L/m$^2$/year (Table 4).

### *Dew zone F– Oman Sea region*

The coastline along the Oman Sea and the strait of Hormuz formed the sixth dew zone, which is also the smallest dew zone in Iran covering only 5% of the grid points (Fig. 5). The overall mean daily dew yield in this zone was about 0.09 L/m$^2$ and the maximum dew yield was about 0.23 L/m$^2$/day (Table 4), which was the highest among all dew zones. This is not surprising because this region benefits from a generous source of moisture (i.e., Oman Sea, Persian Gulf, Arabian Sea, Indian Ocean through the summer monsoon). Observations from the station Bandarabas confirm the presence of a moisture source as the

difference between the temperature and dew point temperature is quite small (about 5°C) and constant throughout the year. However, despite these conditions, the formation of dew was mostly limited to the cold season (i.e., starting in September and ending by March, Fig. 6) and during the warm season (i.e., April–August), dew occurrence was rare (Fig. 6). The reason is likely due to the increase in wind speed (as shown by obseravations at the synoptic station Bandarabas, Fig. 8e)  and the pemperature during summer. In particular, in the warm season, high temperatures leads to the formation of low-pressure

systems (i.e., Gang and Persian Gulf) over the seas, which intensified the hot and humid conditions in the southern coastal region. High humidity results in amplified long-wave radiation downwards, and therefore less radiative cooling. In addition, due to the strong gradient between the low pressure over the Persian Gulf and the high pressure over Saudi Arabia, an intense airflow is stimulated, so that condensation does not occur despite high humidity. Lastly, although this zone had the highest daily dew yield, it does not have the highest yearly yield (i.e., > 80 L/m$^2$) since the frequency of dewy days (~ 150 days) in

this zone is lower than in dew zone A (the Caspian Sea region, Fig. 7

### 3.2.2 Long-term temporal variation in dew formation zones

In order to investigate the long-term (1979-2018) variation of dew formation, we applied the Mann-Kendal trend test (Pohlert, 2016) to the yearly means of dew yield with a confidence level of 95%. Figure 9 shows the statistical significance ($p < 0.05$) of the overall changes in the mean yearly dew yield. The result of this trend analysis showed that in more than 60% of the land

areas in Iran (i.e., mostly dew zones C and F and the northern half of dew zones B and D), dew formation has decreased during the past 40 years. The remaining parts of Iran did not show any significant trend ($\alpha = 5\%$) however, their negative slope (82% of the remained grid points) might be a sign of a future decrease in dew formation for these regions. Such negative trends in dew yield over a wide geographical region could be due to different reasons that control the condensation process. To identify potential causes for the detected decrease in dew formation, we first calculate correlations between the dew formation and

meteorological parameters (temperature, dewpoint temperature, dewpoint depression, relative humidity, wind speed, and cloud

cover, obtained from ERA-Interim) for each dew formation zone (Table 5). Subsequently, we calculate the trend for each of the 6 meteorological variables.

The correlation analysis (i.e., Pearson's correlation) revealed that dew formation in almost all dew zones (i.e., B-F) has a very strong negative correlation (values of -0.93 to -0.95) with temperature, a strong positive correlation with relative humidity (values of 0.88 to 0.98), and a negative correlation with the dewpoint depression (-0.69 to -0.88). In contrast, dew zone A (the Caspian Sea) has weak correlations between dew formation and temperature, relative humidity, and dewpoint temperature indicating dew formation in this region is controlled by different processes. In addition, zone A is the only zone to have a weak and negative correlation between dew formation and cloud cover. These huge differences between dew zone A and other zones are likely due to differences in topography as dew zone A is mainly covered by forests and the behaviour of some climatological variables can be different than the rest areas. A moderate negative correlation between dew formation and wind speed (-0.62) does exist in zone A which may indicate that wind speed is the meteorological parameter with the most influence on dew formation in Zone A.

When the long-term trends are considered, air temperature, which has a negative effect on dew formation, showed a significant positive trend ($p < 0.05$) in all dew zones over the 40 years. The magnitude of these changes for zones A–F was 0.6, 0.6, 0.7, 0.4, 0.3, and 0.3 °C per decade; respectively. Relative humidity (RH) and cloudiness had a positive effect on dew formation (except in Zone A), however, they both had a negative trend over 40 years. The average decrease in relative humidity for dew zones (i.e., A-E) was about 1.5% / decade (Table 5). Therefore, the increase in temperature and decrease in RH and cloudiness can largely explain the decreasing trend in dew yield during the last 4 decades (1979–2018).

### 3.3 Uncertainties in the dew model simulation results

A detailed investigation of the model setup (i.e. input parameter (e.g. emissivity and albedo), wind profile assumption, heat transfer assumptions, etc.) revealed that the dew yield estimation is very sensitive to the heat transfer coefficient. In order to obtain an estimation of the final uncertainties in the model simulation results (i.e. daily dew yield) caused by the heat transfer coefficient, we ran the model with eight different parameterizations of the heat transfer coefficients for four grid points (Table 6) for one year (2000). We selected 4 stations / grid points (red stars in Fig. 1) in different dew zones: Ramsar station in dew zone A, Zanjan in dew zone B, Tabas in dew zone D, and Bandarabas in dew zone F.

Figure 10 shows the daily dew yields estimated using the parameterization by Richards (2009, this study) against the daily dew yields obtained from the seven other parameterizations listed in Table 6. for all 4 grid points considered, the parameterizationss of Beysens (2005) and Watmuff (1997) give the largest estimates of daily dew yields whereas the parmeterizations of Kumar (1997) and Maestre-Valero et al. (2011) give the lowest estimates (Fig 10, Table 6). Figure 10 also demonstrates that the largest differences occur for the very low values of daily dew yield.

The absolute differences in daily dew yields between the parameterizations are calculated in Table 6. At Ramsar, (located in dew zone A, in a forested region), the daily mean dew estimates range from 0.10 to 0.19 L /m$^2$ which is the largest absolute

range of 4 stations. However, the daily dew yield at Ramsar, from all parameterizations, is the largest of all 4 stations (Table 6) and thus this station has the smallest relative difference: the smallest estimate of 0.10 L /m$^2$ is 83% of the value obtained from the Richardson parameterization and the largest estimate of 0.19 L /m$^2$ is 158%. Bandarabas (located in dew zone F, coastal region) also has a large range of daily dew estimates (0.01 to 0.14 L /m$^2$ ) but combined with the much lower daily dew yield (0.04 L /m$^2$ from the Richards parameterization) means this grid point has a much larger relative variation (25 % to

350%) in estimated daily dew yields in comparison to Ramsar. The heat capacity parameterization has a strong impact on the modelled daily dew yields, however the standard deviations of the daily means are also large (Table 6). However, we conclude that the parameterization used in this study – the Richards (2009) formulation - gives estimates that are similar to the average of all methods and are neither much lower nor much higher than the majority of other parameterizations.

In an ideal situation we would compare our model results to observations, however, unfortunately observational data of dew
formation in Iran is not available. Therefore, the accuracy of the modelled dew yields in comparison to observational data cannot be performed for this study. However, Vuollekoski et al. (2015, Section 2.3 and Fig. 2) and Atashi et al. (2021) presented detailed comparisons between results from this dew model and observations in other locations, where experimental dew data was available. The results of these studies revealed that in most cases the model overestimates the dew yield due to some limitations that discussed in section 2.2; however, the cumulative sum of observed and simulated dew yield were found
to agree well after with smoothing down the daily variations.

## 4. Discussion

Iran is a country located in arid and semi-arid regions, which has a growing population and has suffered from water scarcity over the last decades. Therefore, finding renewable sources of water is rapidly becoming a necessity. Dew is one of these
atmospheric resources of water which can be vital especially in more dry conditions.

The average daily dew yield in Iran was in the range of 0.03 – 0.14 L/m$^2$ and the maximum was in the range of 0.29 – 0.52 L/m$^2$/day. Our modelled-based results are largely in agreement with previous observational dew measurement studies conducted in similar climates (i.e., arid and semi-arid, coastal desert, Mediterranean) using planer dew condensers. However, the quantitative estimates of dew formation can differ between stations located within the same climatic zone. For instance,
the reported values for average and maximum daily dew yield for semi-arid Mediterranean climate (similar to dew zone A and some parts of zone B in Iran) was 0.04 and 0.33 L/m$^2$ in Zadar (France; Muselli et al., 2009), 0.09 and 0.48 L/m$^2$ in Komiza (Croatia; Muselli et al., 2009), 0.04 and 0.27 L/m$^2$ in Beirut (Lebanon; Tomaszkiewicz and Abou Najm, 2015), 0.06 – 0.19 and 0.48 L/m$^2$ in a semi-arid coastal area in south-western Madagascar (Hanisch et al., 2015), 0.13 and 0.46 L/m$^2$ in Beiteddine village (Lebanon; Tomaszkiewicz et al., 2017). The coastal desert area (i.e., Zone E and F) can be comparative with the
observed values in Nitzana, Israel (mean: 0.09 L/m$^2$ Kidron, 1999), Dhahan, Saudi Arabia (mean: 0.22 L/m$^2$/day; Gandhisan and Abualhamayel 2005), Panandhro, India (mean: 0.18 and max: 0.56 L/m$^2$/day; Sharan et al., 2011). The average frequency

of dew occurrence in Iran was 102 days, while the average number of rainy days in Iran is 38 days (Kashki and Dadashi Roudbari, 2017), suggesting that dew is more frequent than rain. Furthermore, a comparison between the total amount of harvestable dew water with rainfall in seven different stations in different climate zones in Iran performed by Atashi et al (2019) revealed that in the arid coastal areas in the south and in central desert areas, dew formation could be about 25% of rainfall which is significant ( see Atashi et al., 2019, Section 3.3, Fig. 4 for further details).

Water scarcity is becoming even more serious with global warming and the impacts of climate change on water resources. As such, the dew formation yields calculated in this study showed a significant decreasing trend in the majority of Iran over the last 4 decades. Similar decreases in dew have also been reported in different areas of the world. Xu et al (2015) investigated the effects of global warming on dew variation in a paddy ecosystem in China (The Sanjiang Plain of Heilongjiang Province) over the last 50 years. Their findings showed that with the current rate of change in T and RH, the average daily dew intensity would decline by 0.036 mm/year. They suggested that a warmer and drier climate would lead to a reduction in dew amount because water cannot condense when RH falls below 71%. In another study, Tomaszkiewicz et al., (2016) used the forecast trends in temperature and relative humidity to estimate dew yields under future climatic scenarios for 142 stations in the Mediterranean region during the critical summer months at the end of the century (2080). Their study predicted that dew harvesting may decline (up to 27%) by the end of the century during the dry season.

In closure, it should be noted that a reliable prediction of dew is still a challenge and the model used in this study has also some limitations (i.e., heat (h) and mass (k) transfer coefficient are semi-empirical parameters, spatial and time resolution of gridded data, the "dew collecting" method in the model might be different than the measurement studies, etc.) that tends to overestimate the daily dew yield. However, uncertainty in the results caused by model assumptions is very unlikely to affect the main conclusions of this study. Namely, these uncertainties do not affect the spatial (dew zones) and temporal (seasonal variation) patterns, nor the obtained results for the historical climate change impact on dew yield. Lastly, to obtain more accurate estimates of future dew formation, and thus a robust scientific basis for future water resource plans to be built upon, our dew formation model should be calibrated with actual dew experimental observations in multiple different climates; this is a topic on ongoing work. Finer spatial and temporal data resolution would also help to resolve local variations in microclimates.

## 5. Conclusion

Iran is a relatively dry country with a limited source of water. Water scarcity has been a serious problem over decades, so that, considering renewable resources of water is imperative. Dew is a non-conventional atmospheric source of water that can be vital, particularly in arid and semi-arid climates, where other water resources are rare. Therefore, in this study, we estimated the potential of dew water yield, identified the main dew zones in Iran and investigated the impacts of already detected climate change on dew formation. In order to estimate dew potential, we used an analytical model based on mass and heat balance between a condenser sheet and the atmosphere. Long-term (1979-2018) model simulation results revealed that dew can form

almost everywhere in Iran, even in hyper dry deserts. The average of dew events was ~ 102 days, with the lowest number of dewy days in summer (~7 days) and the highest in winter (~ 45 days). The average daily dew yield was also in the range of 0.01-0.14 $L/m^2$ with the maximum yields in winter (0.23 mm/day). In both dew occurrence and yield, the coastal and mountain parts of Iran had the highest values and interior and eastern areas had the lowest values. Conserning the uncertainty in the model simulation results, the uncertainty analysis with eight different parameterizations of the heat transfer coefficients (dew yield estimation is very sensitive to the heat transfer coefficient) for four grid points (in different dew zones) for one year (2000). The results revealed that the parameterization used in this study – the Richards (2009) formulation - gives estimates that are similar to the average of all methods and are neither much lower nor much higher than the majority of other parameterizations and the largest differences occur for the very low values of daily dew yield.

In order to identify the dew formation zones in Iran, we used a hierarchical agglomerative clustering method which identified 6 distinct dew zones. The geographical variation of the dew formation zones closely matched with the topography and the sources of moisture (e.g., nearby sea areas) in Iran. Zone A (i.e., Caspian Sea) had the highest overall mean daily dew occurrence (~ 330 days) and yield (0.14 $L/m^2$), and Zone D (i.e., Lut desert zone), had the lowest dew events (~ 15 days) and yields (0.03 $L/m^2$).

The Mann-Kendal trend test revealed a significance ($p < 0.05$) negative trend in the yearly dew yield in the majority of Iran during the last 4 decades (1979- 2018). This reduction in dew was mainly the result of increases in air temperature and decreases in relative humidity which are key factors in dew formation.

**Data availability**

The model and data used in this study are publicly available and can be accessed as follows:

- The program source code, written in Python and Cython is available at https://github.com/vuolleko/dew_collection/.
- The meteorological input data using The European Centre for Medium-Range Weather Forecasts (ECMWF) reanalysis and forecast fields (ERA-Interim): https://apps.ecmwf.int/datasets/data/interim-full-daily/levtype=sfc/.

**Author contribution**

NA: Performed the model code, analysis, visualization, and writing the original paper. TH; TV; DR; and MK: Supervision, funding the research. M.A.Z; A.R and V.A.S: Editing the paper; HV: developed the original model code.

**Competing interests**

The authors declare that they have no conflict of interest.

**Acknowledgment**

The University of Isfahan is acknowledged to facilitate the research visit abroad for graduate students. The Ministry of Science, Research, and Technology supported Miss Nahid Atashi to visit the University of Helsinki, Institute for Atmospheric and Earth System Research (UHEL- INAR). The University of Helsinki hosted Miss Atashi during the first twelve months of her research

visit working on dew yield potential modelling. This visit was also under the Academy of Finland Center of Excellence programme (CoE-ATM, grant no. 307331) and Academy Professor projects (312571 and 282842).

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

 **Table 1.** Dew yield from plane radiative condensers in various field campaigns and models.

| Sampling site | Dew events | Study period | Mean volume [L/m2/day] | Max volume [L/m2/ day] | Observed/ Modeled | Reference |
|---|---|---|---|---|---|---|
| Fayetteville, AR (USA) | 107 | Jul 1989–July 1990 | 0.15 | - | Obs | Wagner et al. (1992) |
| Dodoma (Tanzania) | - | 30 nights | 0.04 | - | Obs | Nilsson (1994) |
| Kungsbacka (Sweden) | 11 | 14 Aug–01 Sept 1993 | 0.145 | 0.21 | Obs | Nilsson, (1996) |
| Dodoma (Tanzania) | 21 | Nov 1993 | 0.057 | 0.08 | Obs | Nilsson, (1996) |
| Dodoma (Tanzania) | 147 | 25 Aug 1994–4 Feb 1995 | 0.05 | 0.24 | Obs/Mod | Vargas et al. (1998) |
| Sde Boqer (Israel) | 34 | Aug–Nov 1992 | 0.2/dew & fog | - | Obs | Kidron (1999) |
| Har Harif (Israel) | 21 | Aug–Nov 1992 | 0.3/dew & fog | - | Obs | Kidron (1999) |
| Dayalbagh (India) | - | 15 Dec–15 Feb | 0.59 | 1.38 | Obs | Khare et al. (2000) |
| Ajaccio (France) | 214 | 22 July 2000–11 Sept 2001 | 0.12 | 0.38 | Obs | Muselli et al. (2002) |
| Osaka (Japan) | 16 | No info | 0.14 | - | Obs | Takenaka et al. (2003) |
| Grenoble (France) | 109 | 25 Nov1999– 23 Jan 2001 | 0.036 | - | Obs | Beysens et al. (2003) |
| Zadar (Croatia) | 87 | 21 July 2003–31 May 2004 | 0.15 | - | Obs | Mileta et al. (2004) |
| Jerusalem (Israel) | 176 | 01 June 2003–31 May 2004 | 0.188 | ~ 0.50 | Obs | Berkowicz et al. (2004) |
| Komizˇa (Croatia) | 76 | 24 June 2003–26 April 2004 | 0.08 | - | Obs | Mileta et al. (2004) |
| Bordeaux (France) | 211 | 14 Aug 1999–23 Jan 2001 | 0.046 | - | Obs/Mod | Beysens et al. (2005) |
| Dhahran (Saudii Arabia) | - | | 0.22 | - | Obs/Mod | Gandhisan and Abualhamayel (2005) |
| Brive-la-Gaillarde (France) | 275 | 01 Jan–31 Dec 2000 | 0.115 | <0.475 | Obs | Beysens et al. (2006a) |
| Ajaccio (France) | - | 10 Dec 2001–10 Dec 2003 | ~ 0.106 | ~ 0.332 | Obs | Muselli et al. (2006a) |
| Bordeaux (France) | 110 | 15 Jan 2002–14 Jan 2003 | - | ~ 0.22 | Obs | Beysens et al. (2006b) |
| Jerusalem (Israel) | 554 | 2003–2006 | 0.199 | ~ 0.60 | Obs | Berkowicz et al. (2007) |
| Kothara (India) | - | 01 Oct 2004–31 May 2005 | 0.098 | 0.24 | Obs | Sharan et al. (2007) |
| Central Netherlands | - | Dec 2003–May 2005 | 0.10 | - | Obs | Jacobs et al. (2008) |
| Tahiti | 151 | 16 May–14 Oct 2005 | 0.068 | 0.22 | Obs | Clus et al. (2008) |
| Tikehau | 109 | 21 June–07 Oct 2005 | 0.102 | 0.23 | Obs | Clus et al. (2008) |
| Komizˇa (Croatia) | 263 | 07 Jan 2003–31 Oct 2006 | 0.108 | 0.592 | Obs | Muselli et al. (2009) |
| Zadar (Croatia) | 484 | 07 Jan 2003–31 Oct 2006 | 0.138 | 0.406 | Obs | Muselli et al. (2009) |
| South–West Morocco | 178 | 01 May 2007–30 April 2008 | 0.106 | - | Obs | Lekouch et al. (2010a) |
| Wrocław (Poland) | 421 | 05 Oct 2007–07 March 2010 | 0.103 | 0.354 | Obs | Sobik et al. (2010) |
| Sudetes (Poland) | 55 | 21 June 2009–16 Jan 2010 | 0.190 | 0.452 | Obs | Sobik et al. (2010) |
| Cartagena (Spain) | 175 | May 2009–May 2010 | 0.105 | - | Obs | Maestre-Valero et al. (2011) |
| Panandhro (India) | 69 | 07 Feb 2004–25Feb 2006 | 0.189 | - | Obs | Sharan et al. (2011) |
| Mirleft (Morocco) | 178 | 01 May 2007–30 April 2008 | 0.106 | - | Obs/Mod | Lekouch et al. (2012) |
| Id Ouasskssou (Morocco) | 187 | 01 May 2007–30 April 2008 | 0.202 | - | Obs | Lekouch et al. (2012) |
| Wroclaw (Poland) | 19 | April–Sep 2009 | 0.179 | - | Obs | Galek et al. (2012) |
| Sde Boqer (Israel) | 29 | during the fall of 1992 | 0.21 | - | Obs | Kidron & Starinsky (2012) |
| Taklimakan Desert (China) | 104 | June–October 2011 | ~0.12 | - | Obs | Hao et al. (2012) |
| Idouasskssou (Morocco) | 137 | 15 Dec 2008–31 July 2009 | 0.158 | - | Obs | Clus et al. (2013) |
| Adelaide Hills (Australia) | 14 | 24 April–23 May 2009 | 0.225 | - | Obs/Mod | Guan et al. (2014) |
| Krakow (Poland) | 79 | May–Oct 2009 | 0.11 | - | Obs | Muskala et al. (2015) |
| Gaik-Brzezowa (Poland) | 80 | May–Oct 2009 | 0.19 | - | Obs | Muskala et al. (2015) |
| Developed in Finland | - | 1979–2012 | - | - | Glob Mod | Vuollekoski et al. (2015) |
| coastal south-western (Madagascar) | - | April 2013–Sep 2014 | 0.06–0.19 | 0.48 | Obs | Hanisch et al. (2015) |
| Developed in France | - | - | - | - | Glob Mod | Beysens (2016) |
| Baku (Azerbaijan) | 118 | April 2010–March 2011 | 0.13 | 0.52 | Obs | Meuniera and Beysens (2016) |
| Mexico City (Mexico) | - | 22 Dec 2011–21 Mar 2012 | 0.0317 | - | Obs | Arias-Torres & Flores-Prieto (2016) |
| Paris (France) | 63 | April 2011–Mar 2012 | 0.055 | - | Obs | Beysens et al. (2017) |
| Beiteddine (Lebanon) | 123 | 2013–2014 growing seasons | 0.13 | 0.46 | Obs | Tomaszkiewicz et al. (2017) |
| Maktau (Kenya) | - | April 2016–Mar 2017 | 0.067 | > 0.15 mm | Obs/Mod | Tuure et al. (2019) |

**Table 2.** Description of the dew formation model by listing the terms in Eq. (1).

| Term | Unit | Description |
|---|---|---|
| $dT_c/dt$ | K s$^{-1}$ | Change rate of the condenser temperature |
| $T_c$ | K | Temperature of the condenser |
| $t$ | s | Time. Here the time step in the model was 10 s |
| $C_c$ | J kg$^{-1}$ K$^{-1}$ | Specific heat capacity of the condenser. For low-density polyethylene (LDPE) and polymethylmethacrylate (PMMA) it is 2300 J kg$^{-1}$ k$^{-1}$ |
| $C_i$ | J kg$^{-1}$ K$^{-1}$ | Specific heat capacity of ice (2110 J kg$^{-1}$ k$^{-1}$) |
| $C_w$ | J kg$^{-1}$ K$^{-1}$ | Specific heat capacity of water (4181.3 J kg$^{-1}$ k$^{-1}$) |
| $m_c$ | kg | Mass of the condenser given by $m_c = \rho_c S_c \delta_c$<br>where $\rho_c$, $S_c$, and $\delta_c$ are the density (here it is 920 kg m$^{-3}$), surface area (here it is 1 m$^2$), and thickness of the condenser (here it is 0.39 mm) |
| $m_i$ | kg | Mass of ice |
| $m_v$ | kg | Mass of water, representing the cumulative mass of water that has |
| $P_{rad}$ | W | Heat exchange due to incoming and outgoing radiation<br>$P_{rad} = (1 - a) S_c R_{sw} + \varepsilon_c S_c R_{lw} - P_c$<br>where $a$ is the condenser short-wave albedo (here it is 0.84), $S_c$ is the condenser surface area (here it is 1 m$^2$), $\varepsilon_c$ is the emissivity of the condenser (here it is 0.94), and $P_c$ is the outgoing radiative power, is given by Stefan-Boltzmann low: $P_c = S_c \varepsilon_c \sigma T_c^4$, $\sigma$ is Stephan-Boltzmann constant (5.67×10$^{-8}$ W m$^{-2}$ K$^{-4}$), $T_c$ [K] is the temperature of the condenser, and $R_{sw}$ and $R_{lw}$ [W m$^{-2}$] are the incoming short-wave radiation (i.e., surface solar radiation downwards) and incoming long-wave radiation (i.e., surface thermal radiation downwards). |
| $P_{cond}$ | W | Conductive heat exchange between the condenser surface and the ground. For simplicity, we assumed that the condenser is perfectly insulated from the ground; i.e., $P_{cond} = 0$ |
| $P_{conv}$ | W | Convective heat exchange<br>$P_{conv} = S_c (T_a - T_c) h$<br>where $S_c$ is the condenser surface area (here it is 1 m$^2$), $T_a$ [K] is the ambient temperature at 2 meters from the ground, $T_c$ [K] is the temperature of the condenser, and $h$ [W m$^{-2}$ K$^{-1}$] is the heat transfer coefficient that is estimated based on a semi-empirical equation (Richards, 2009)<br>$h = 5.9 + 4.1\ WS\ (511 + 294) / (511 + T_a)$<br>and here $WS$ [m s$^{-1}$] is the prevailing horizontal wind speed at 2 meters from the ground. |
| $P_{lat}$ | W | Latent heat released by the condensation or desublimation of water<br>$$P_{lat} = \begin{cases} L_{vw} \dfrac{dm_w}{dt} & T_c > 0\ ^oC \\ L_{vi} \dfrac{dm_i}{dt} & T_c < 0\ ^oC \end{cases}$$<br>where $L_{vw}$ [J kg$^{-1}$] is the specific latent heat of water vaporization and and $L_{vi}$ [J kg$^{-1}$] is specific latent heat of water desublimation. Here, $dm_w/dt$ is the change rate of water whereas $dm_i/dt$ is the change rate of ice |


**Table 3.** A list of nomenclature.

| Parameter | Unit | Description |
|---|---|---|
| $\alpha$ | -- | Albedo of condenser sheet |
| $C_a$ | J kg$^{-1}$ K$^{-1}$ | Specific heat capacity of air |
| $C_c$ | J kg$^{-1}$ K$^{-1}$ | Specific heat capacity of the condenser |
| $C_i$ | J kg$^{-1}$ K$^{-1}$ | Specific heat capacity of ice |
| $C_w$ | J kg$^{-1}$ K$^{-1}$ | Specific heat capacity of water |
| $DP$ | K | Dew point temperature |
| $h$ | W K$^{-1}$ m$^{-2}$ | Heat transfer coefficient |
| $k$ | Per s$^{-1}$ | Mass transfer coefficient |
| $L_{vi}$ | J kg$^{-1}$ | Specific latent heat of desublimation for water |
| $L_{vw}$ | J kg$^{-1}$ | Specific latent heat of vaporization for water |
| $L_{wi}$ | J kg$^{-1}$ | Latent heat of fusion |
| $m_c$ | kg | Mass of the condenser |
| $m_i$ | kg | Mass of ice |
| $m_w$ | kg | Mass of water |
| $p$ | Pa | Atmospheric air pressure |
| $p_c$ | Pa | Vapor pressure over condenser |
| $p_{sat}$ | Pa | Saturation pressure of water |
| $P_{cond}$ | W | Conductive heat exchange between the condenser surface and the ground |
| $P_{conv}$ | W | Convective heat exchange |
| $P_{lat}$ | W | Latent heat released by the condensation or desublimation of water |
| $P_{rad}$ | W | Heat exchange due to incoming and outgoing radiation |
| $R_{lw}$ | W m$^2$ | Surface thermal radiation downwards |
| $R_{sw}$ | W m$^2$ | Surface solar radiation downwards |
| $S_c$ | m$^2$ | Surface area of condenser |
| $T_a$ | K | Ambient temperature at 2 meters |
| $T_c$ | K | Temperature of the condenser |
| $U_{10}$ | m s$^{-1}$ | Horizontal wind speed component at 10 meters |
| $V_{10}$ | m s$^{-1}$ | Horizontal wind speed component at 10 meters |
| $WS$ | m s$^{-1}$ | Prevailing horizontal wind speed at 2 meters |
| $z_0$ | m | Surface roughness |
| $\delta_c$ | mm | Condenser sheet thickness |
| $\varepsilon_c$ | -- | Emissivity of condenser sheet |
| $\gamma$ | Pa K$^{-1}$ | Psychrometric constant |
| $\sigma$ | W m$^{-2}$ k$^{-4}$ | Stefan-Boltzmann constant |

**Table 4.** Dew formation zones and their climate features (i.e., mean (min–max) values for meteorological parameters (T, $T_d$, RH)) as well as statistical analysis for overall mean daily cumulative dew yield (i.e., std, 25, 50, 75th and 99th percentile as daily max as well as yearly max dew yield).

| | Zone A | Zone B | Zone C | Zone D | Zone E | Zone F |
|---|---|---|---|---|---|---|
| $T_{mean}$ [°C] | 12 (-1–23) | 12 (-1–26) | 17 (3–31) | 20 (7–33) | 22 (9–35) | 27 (16–36) |
| $T_{d\ mean}$ [°C] | 5 (-5–14) | 1 (-6–6) | 1 (-5–6) | 0 (-4–4) | 6 (2–9) | 10 (3–17) |
| $RH_{mean}$ [%] | 69 (58–81) | 52 (27–77) | 40 (21–67) | 30 (15–56) | 37 (15–66) | 39 (25–54) |
| Mean dew yield ± std [L/m$^2$/day] | 0.14 ± 0.01 | 0.08 ± 0.04 | 0.05 ± 0.05 | 0.03 ± 0.03 | 0.06 ± 0.04 | 0.09 ± 0.06 |
| 25 % [L/m$^2$] | 0.08 | 0.04 | 0.02 | 0.01 | 0.03 | 0.03 |
| Median [L/m$^2$] | 0.13 | 0.07 | 0.04 | 0.02 | 0.05 | 0.07 |
| 75% [L/m$^2$] | 0.19 | 0.11 | 0.06 | 0.04 | 0.08 | 0.13 |
| 99% [L/m$^2$] | 0.32 | 0.24 | 0.17 | 0.14 | 0.2 | 0.29 |
| Mean [L/ m$^2$/year] | 53 | 30 | 18 | 12 | 24 | 34 |
| Max [L/ m$^2$/year] | > 100 | > 60 | < 50 | < 45 | > 70 | > 80 |


**Table 5**. Correlation between long-term mean daily dew yield and meteorological parameters obtained from ERA Interim for the time period 1979-2018 and Sen's trend slope in the meteorological variables per decades (i.e.,10 years).

| Zone | | T [°C] | Td [°C] | T-Td [°C] | RH [%] | WS [m/sec] | Cloud cover [%] |
|---|---|---|---|---|---|---|---|
| Zone A | Correlation | 0.25 | 0.28 | -0.15 | -0.18 | -0.62 | -0.24 |
| | Trend slope | **0.6*** | 0.1 | **0.4*** | **-1.6*** | **0.02*** | -0.01 |
| Zone B | Correlation | -0.93 | -0.75 | -0.97 | 0.95 | -0.67 | 0.93 |
| | Trend slope | **0.6*** | -0.09 | **0.6*** | **-2*** | **0.02*** | **-0.01*** |
| Zone C | Correlation | -0.96 | -0.88 | -0.96 | 0.98 | -0.75 | 0.84 |
| | Trend slope | **0.7*** | -0.4 | **1*** | **-2.5*** | **-0.04*** | **-0.01*** |
| Zone D | Correlation | -0.94 | -0.74 | -0.95 | 0.98 | -0.37 | 0.74 |
| | Trend slope | **0.4*** | **-0.04*** | **0.4*** | -0.8 | -0.001 | 0 |
| Zone E | Correlation | -0.95 | -0.69 | -0.94 | 0.97 | -0.67 | 0.84 |
| | Trend slope | **0.3*** | **0.4*** | 0.01 | **-0.4*** | **0.05*** | 0 |
| Zone F | Correlation | -0.95 | -0.75 | -0.94 | 0.88 | -0.53 | 0.42 |
| | Trend slope | **0.3*** | **0.4*** | -0.1 | 0.1 | **0.02*** | 0 |

*Values with star indicate a statistically significant trend in ($p<0.05$).

**Table 6.** A selection of 8 various parameterizations for the heat transfer coefficient, the mean±std daily dew yield and [the mean differences between daily cumulative] (L/m$^2$/day; negative values indicate the underestimated dew yield relative to the Richard's parameterization and positive values indicate the overestimated dew yeild) dew yield caused by each parameterization in 4 selected stations compare to the used coefficient in this study (i.e. Richards (2009)). The first three are
studies on dew formation. Here, $u$ and $T_a$ are the horizontal wind speed and air temperature at 2m height, and $L$ is the characteristic length of the condenser (e.g. 1 m).

| source | Parameterization | Ramsar | Zanjan | Tabas | Bandarabas |
|---|---|---|---|---|---|
| Richards (2009); this study | $h = 5.9 + 4.1u\left(\dfrac{511 + 294}{511 + T_a}\right)$ | 0.12±0.12 | 0.05±0.07 | 0.06±0.09 | 0.04±0.07 |
| Beysens et al. (2005) | $h = 4\sqrt{u/L}$ | 0.19±0.10 [0.07] | 0.09±0.07 [0.04] | 0.12±0.11 [0.06] | 0.14±0.12 [0.11] |
| Maestre-Valero et al. (2011) | $h = 7.6 + 6.6u\left(\dfrac{511 + 294}{511 + T_a}\right)$ | 0.10±0.11 [-0.02] | 0.03±0.06 [-0.02] | 0.04±0.07 [-0.02] | 0.01±0.04 [-0.03] |
| Jürges (1924) | $h = 5.7 + 3.8u$ | 0.14±0.12 [0.02] | 0.06±0.07 [0.01] | 0.07±0.10 [0.01] | 0.04±0.07 [0.00] |
| Watmuff et al. (1977) | $h = 2.8 + 3u$ | 0.18±0.11 [0.06] | 0.08±0.07 [0.03] | 0.09±0.11 [0.03] | 0.10±0.11 [0.06] |
| Test et al. (1981) | $h = 8.55 + 2.56u$ | 0.13±0.11 [0.01] | 0.05±0.07 [0.00] | 0.06±0.10 [0.00] | 0.03±0.06 [-0.01] |
| Kumar et al. (1997) | $h = 10.03 + 4.687u$ | 0.10±0.11 [-0.02] | 0.10±0.11 [0.05] | 0.04±0.08 [-0.02] | 0.01±0.04 [-0.03] |
| Sharples and Charlesworth (1998) | $h = 9.4\sqrt{u}$ | 0.15±0.12 [0.03] | 0.06±0.07 [0.01] | 0.07±0.10 [0.01] | 0.04±0.07 [0.00] |

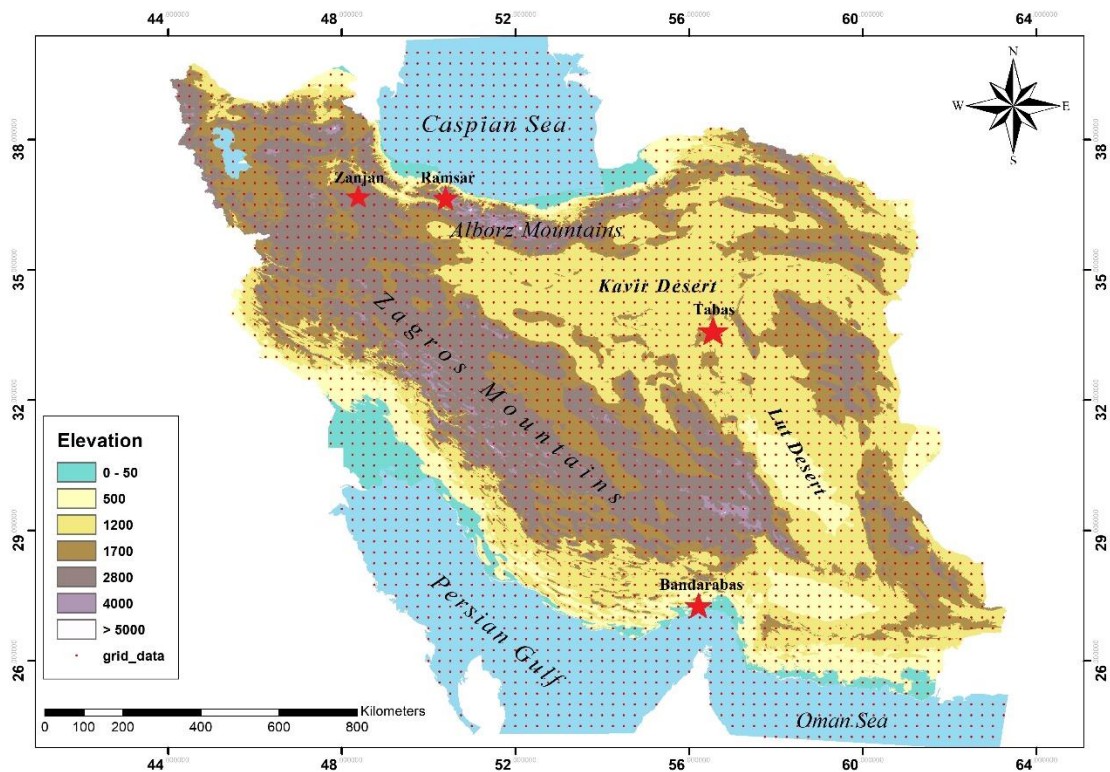

**Figure 1.** A map of Iran illustrating the geographical topography and the domain of the grid points used in the model simulation. Red stars indicate the selected stations for uncertainty analysis.

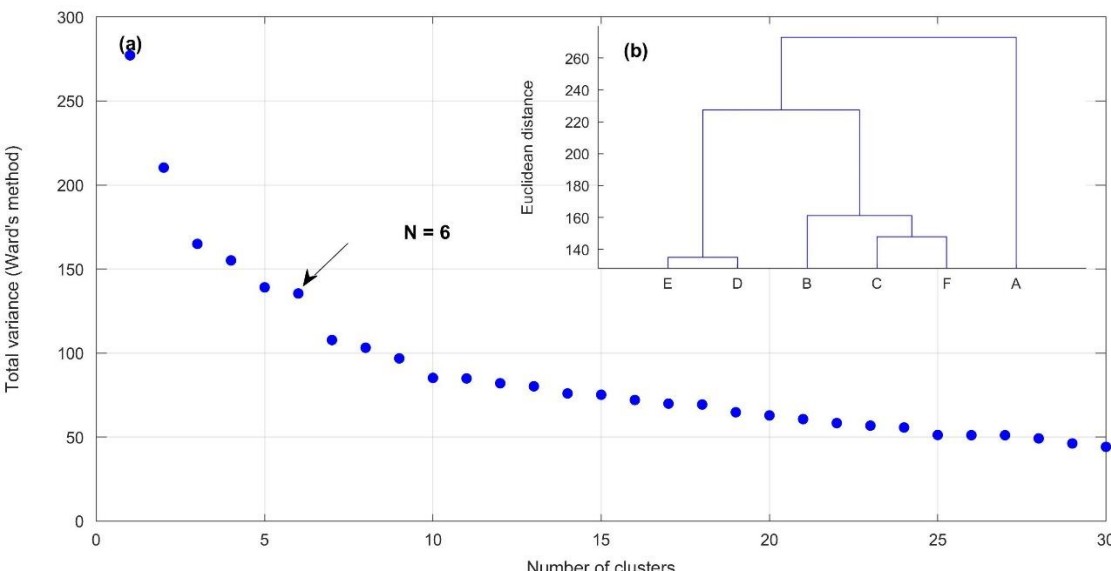

**Figure 2**. (a) Distance level at which two clusters are merged as function of the number of clusters result of the Ward linkage method applied to daily dew yield data from 1979-2018. N is the optimal number of clusters has been chosen for this study and (b) dendrogram of 6 clusters.


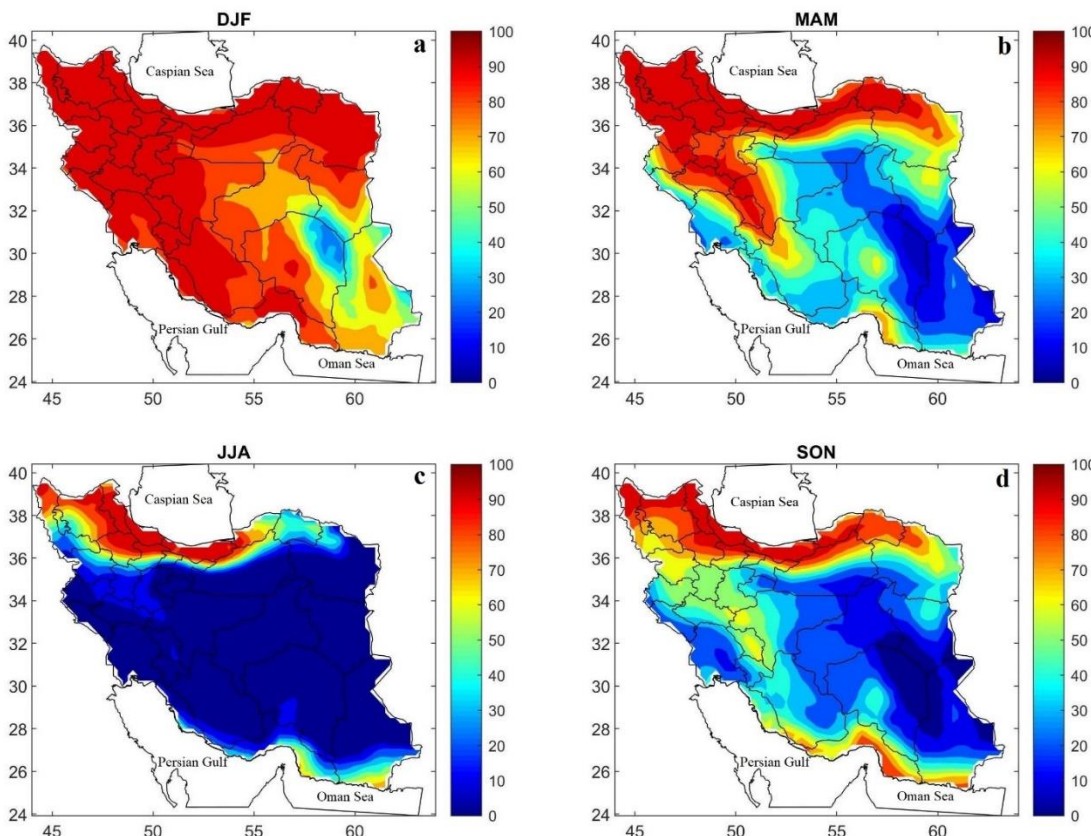

**Figure 3.** Frequency of dew occurrence as fraction of days presented as an overall seasonal mean during 1979–2018. (a) winter (December, January, and February), (b) spring (March, April, and May), (c) summer (June, July, and August), and (d) autumn (September, October, and November).

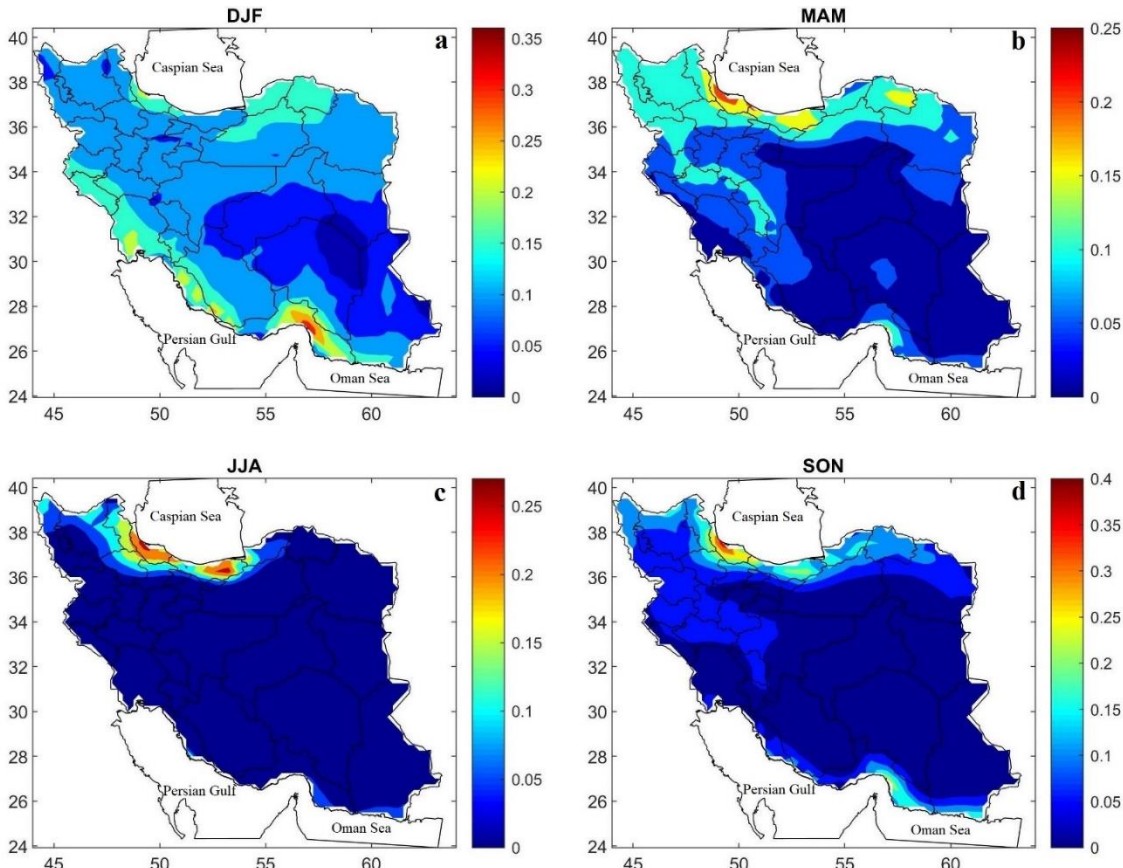

**Figure 4.** Cumulative dew yield [L/m²/day] presented as an overall seasonal mean during 1979–2018. (a) winter (December, January, and February), (b) spring (March, April, and May), (c) summer (June, July, and August), and (d) autumn (September, October, and November).

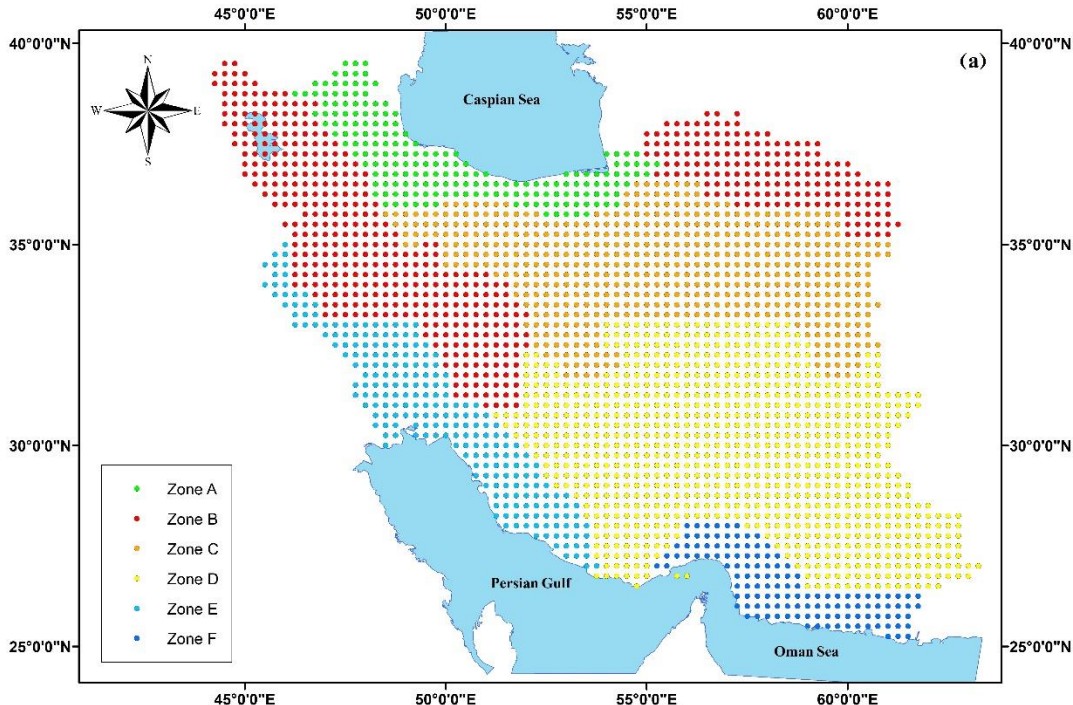

**Figure 5.** Dew formation zones based on the cluster analysis of the daily cumulative dew yield during 1979–2018.


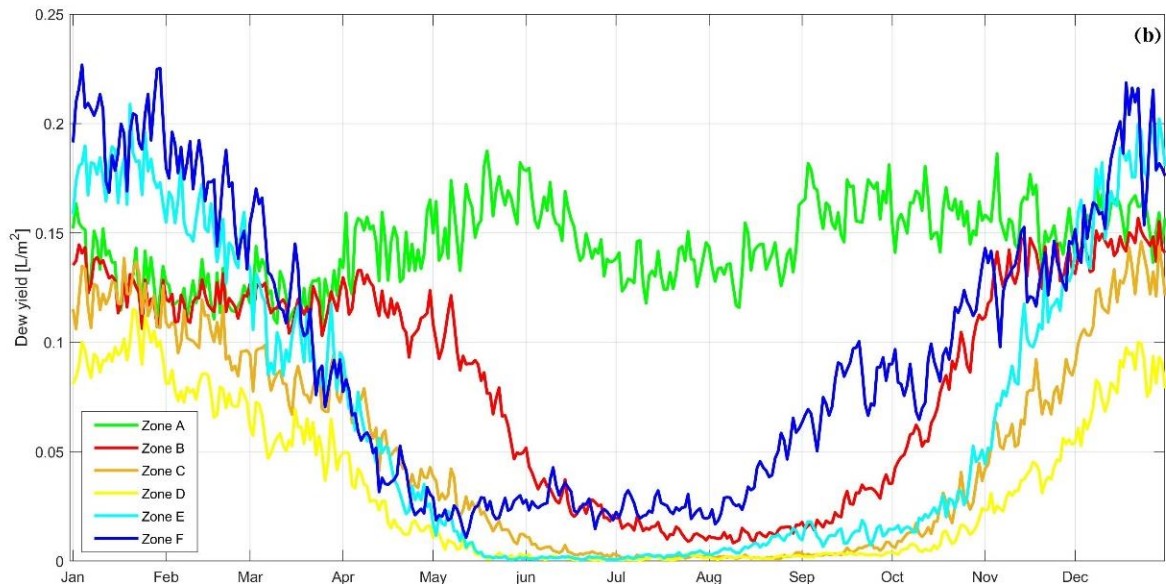

**Figure 6.** Long-term mean seasonal variation of the cumulative daily dew yield. Note that color coding on this figure is the same and corresponds to the dew formation zones on Figure 5: (Green) dew zone A (Caspian Sea), (red) Zone B (Zagros region), (orange) Zone C (Central Iran), (yellow) Zone D (Lut desert), (light blue) Persian Gulf zone, (dark blue) Oman Sea zone. The percentiles (i.e. 25%, median, 75% and 99%) of daily dew yields as average for each cluster are presented in Table 4.

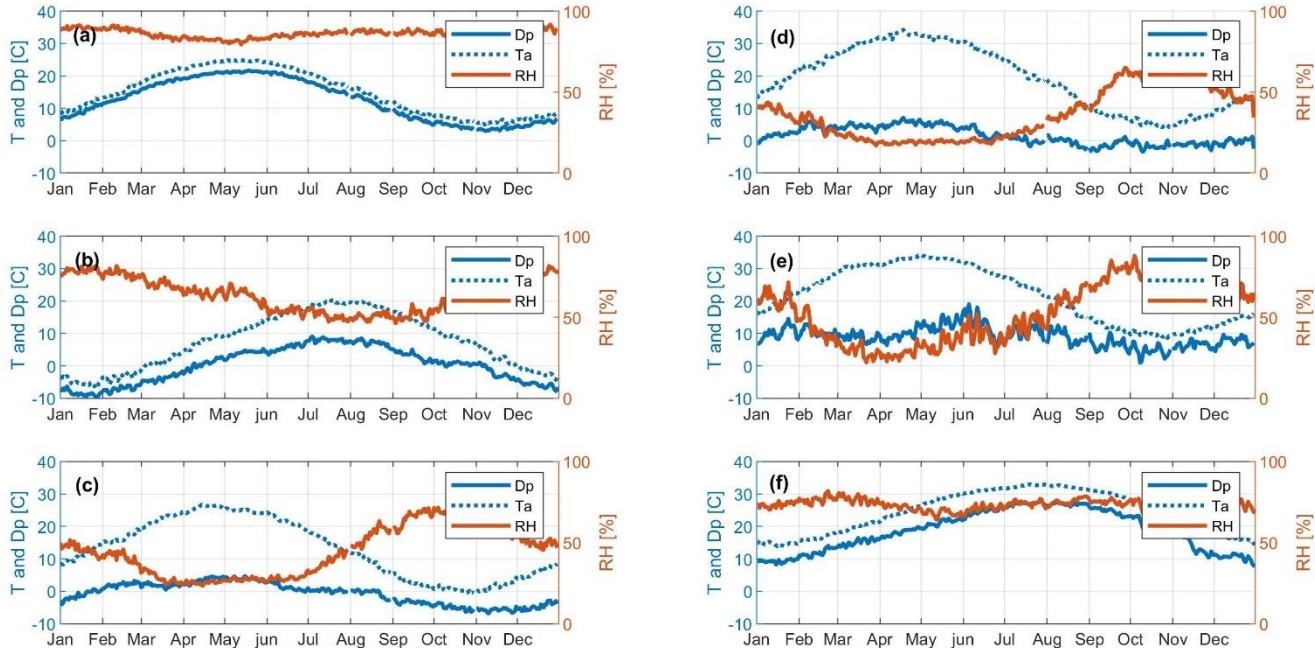

**Figure 7.** Night time (i.e. 18:00, 21:00, 00:00, 03:00) long term mean (1980-2010; 30 years) of dew point temperature (Dp) temperature
(Ta), and relative humidity in six selected stations located in dew zone A-F. (a) Ramsar (dew zone A); (b) Zanjan (dew zone B); (c) Isfahan
(dew zone C); (d) Tabas (dew zone D); (e) Ahvaz (dew zone E), and (f) Bandarabas (dew zone F). Data were obtained from the
meteorological organization of Iran.

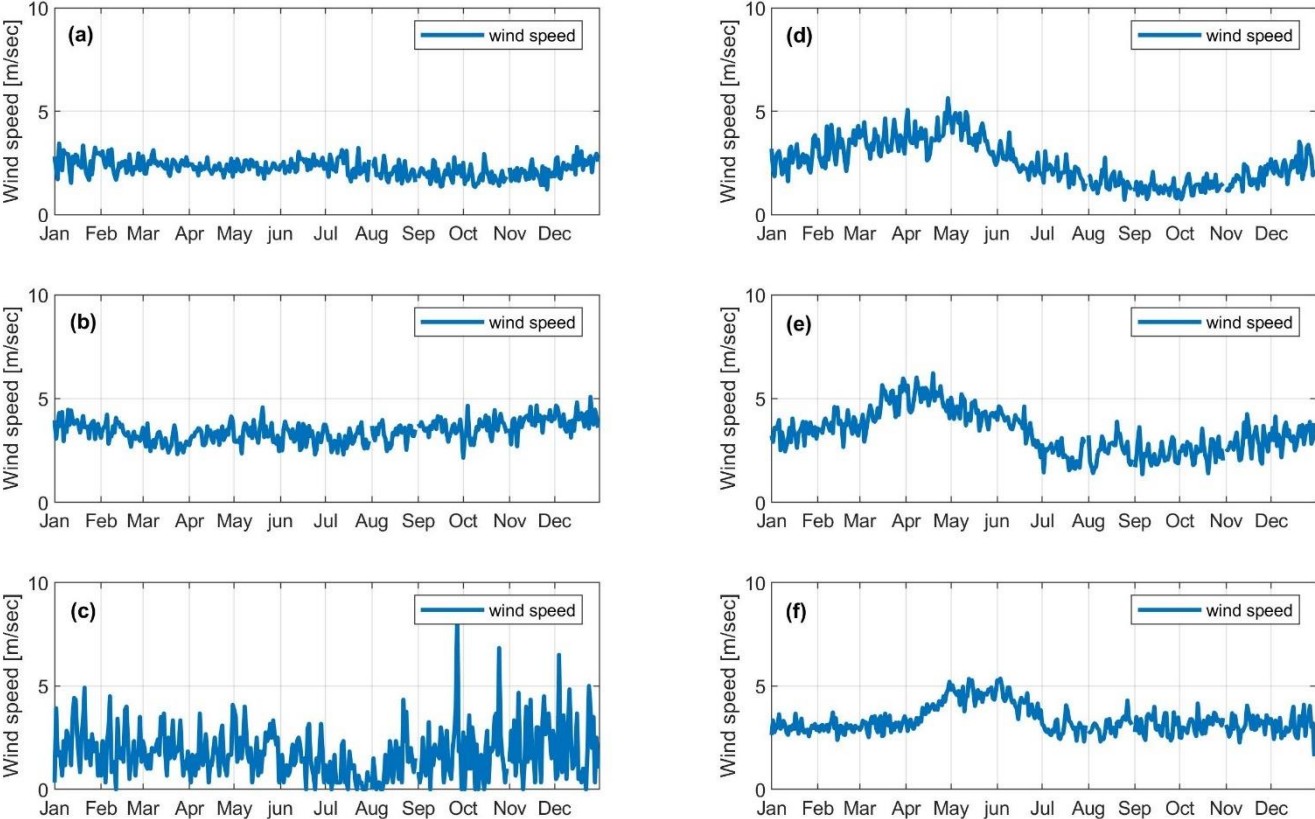

**Figure 8.** Night time **(i.e. 18:00, 21:00, 00:00, 03:00)** long term mean (1980-2010; 30 years) of wind speed in six selected stations that are located in dew zone A-F. (a) Ramsar (dew zone A); (b) Zanjan (dew zone B); (c) Isfahan (dew zone C); (d) Tabas (dew zone D); (e) Ahvaz (dew zone E), and (f) Bandarabas (dew zone F). Data were obtained from the meteorological organization of Iran.

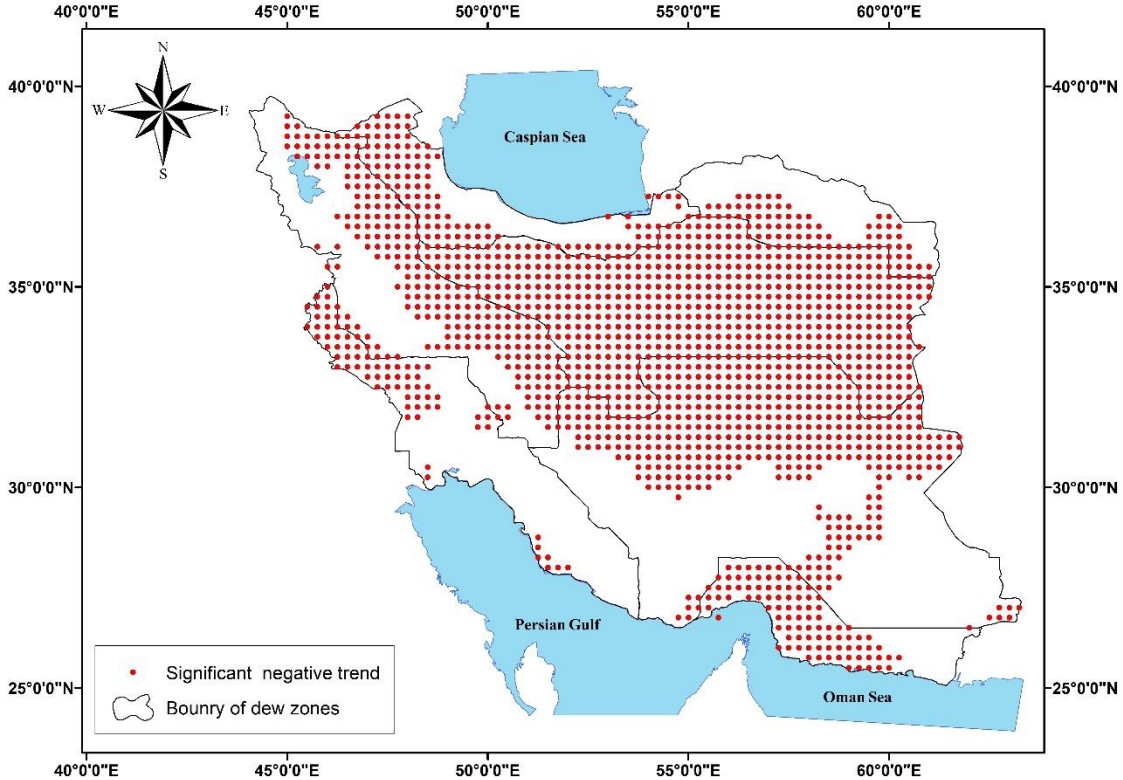

**Figure 9.** Mann-Kendal trend test on mean yearly dew yield over the years 1979- 2018 as predicted by Sen's slope estimator. Only locations with a statistically significant trend (p < 0.05) are shown. Red points present locations with negative trend, regardless their decreased values and the white parts did not show any significant trend at (p < 0.05).

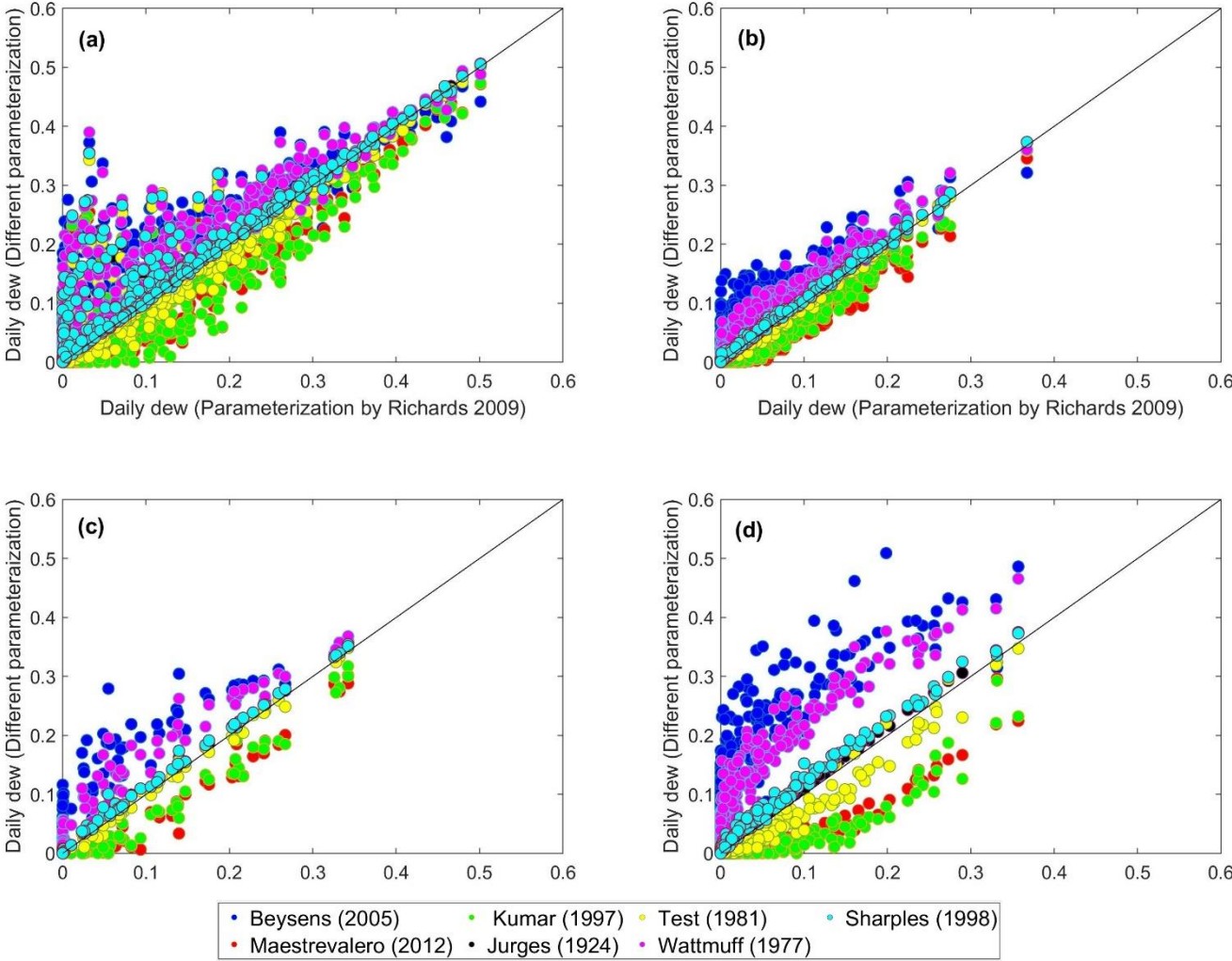

**Figure 10.** Scatter plots of daily dew yield obtained from parameterization by Richards (2009) against 7 different
parameterizations listed in Table 6 for 4 stations in different areas of Iran. (a) Ramsar (forest and coastal area in the north); (b)
Zanjan (mountains area); (c) Tabas (desert area in central Iran), and (d) Bandarabas (arid coastal area in the south). The black
lines are the 1:1 relaithionship.