# Peer review of "Delineation of Dew Formation Zones in Iran Using Long-Term Model Simulations and Cluster Analysis"

_Hydrology and Earth System Sciences, 2021_

## Author Response (AR1)

**#Referee Comment on hess-2021-54**
Al-Kuisi Mustafa

**The manuscript entitled "Delineation of Dew Formation Zones in Iran Using Long-Term Model Simulations and Cluster Analysis" presents valuable results about the dew potential in Iran during a long-term period (40 Years) that can be utilized for strategic planning to harvest due water as an alternate source of water that can be used for agricultural purposes. In general, the manuscript is well written, well structured, and technically sound. I recommend for publication in the Print-Form in Hydrology and Earth System Sciences after minor revisions.**

We thank the reviewer for their constructive comments on our submitted manuscript. We have copied the comments of reviewer here in red and include our response to each individual comment after "Response"

**Specific Comments:**

- **The model input consists of an extensive meteorological data-base that is transformed and processed to be usable according to the model setup. However, equation 1, which is related to the scaling of the wind speed from 10 meters to 2 meters, has limitation that need to be addressed in brief.**

Response: The following references have been added in the revised manuscript clarifying the use of logarithmic wind profile and its limitations under different stratification conditions (line 109).

- Oke, T. R. (2002). Boundary layer climates. Routledge.
- Optis, M., Monahan, A., & Bosveld, F. C. (2016). Limitations and breakdown of Monin–Obukhov similarity theory for wind profile extrapolation under stable stratification. Wind Energy, 19(6), 1053-1072. DOI: 10.1002/we.1883.
- Riou, C. (1984). Simplified calculation of the zero-plane displacement from wind-speed profiles. Journal of Hydrology, 69(1-4), 351-357.
- Holtslag, A. A. M. (1984). Estimates of diabatic wind speed profiles from near-surface weather observations. Boundary-Layer Meteorology, 29(3), 225-250.
- Petersen, E. L., Mortensen, N. G., Landberg, L., Højstrup, J., & Frank, H. P. (1998). Wind power meteorology. Part I: Climate and turbulence. Wind Energy: An International Journal for Progress and Applications in Wind Power Conversion Technology, 1(S1), 25-45.

- **The cluster analysis is a powerful tool to identify the principal groups in the model results. The authors used this method in a brilliant way to identify the dew formation zones and match them with the climate zones in Iran. The authors used Ward's minimum variance method (Ward, 1963) to determine the number of groups (clusters) to define the dew formation zones as shown in Figure 5 and led to six zones. Why six zones were the optimal number? Is it possible to reduce the number of zones?**

Response: In the revised manuscript some brief comments addressing the reasons of choosing 6 clusters has been added (lines 200-202).

- **As a semi-arid climate in the majority of Iran, it is valuable to make a brief comparison between the total amount harvestable dew water and compare it with the total amount of rainfall over Iran. Here, the comparison should not be for the total amount but rather to compare with rainfall depth. The comparison is vital to show that the harvestable dew yield is close to the rainfall depth in some regions.**

Response: A short discussion on comparison between the total amount of harvestable dew water with rainfall based on our previous publication (https://doi.org/10.3390/w11122463) has been added in the revised manuscript (lines 422-425).

- **From the model simulation it looks clear that the regions that received due potential are actually the same ones received the heights rainfall during the year and among Iran. How can this redistribution of the harvestable dew water can enhance to replace the scarcity of water in other regions in Iran such**

**the central and eastern parts that receive the lowest amounts of rainfall and the least potential for dew formation? This is only from a strategic point view to utilize the importance of this study.**

Response: This is true. The areas with high dew potential are the same areas with high precipitation. However, as we highlighted in lines 421-422, the frequency of dew days is about 3 times of the rainy days. This is also true within arid areas in the central and eastern parts of country.

- **It is known that Iran is not the only semi-arid country in the region. How the authors can elaborate the results to be utilized in other similar countries like middle east countries and Afghanistan, Iraq, Syria, Jordan and Gulf countries.**

Response: The importance of the model in that it is applicable all over the world using available grided data has been highlighted in the revised manuscript (lines 160-162).

**Afterall, this kind of modelling studies is needed and recommended in the region. This study has revealed valuable information and understanding about the potential of dew harvesting in such semi-arid regions.**
Response: Thank you for pointing out the importance of this study and modelling tool.

**Comment on hess-2021-54**
**Anonymous Referee #1**

The paper " Delineation of Dew Formation Zones in Iran Using Long-Term Model Simulations and Cluster Analysis" by Nahid Atashi et al., reports an interesting study concerning a simulation of dew yield in Iran based on meteorological data over a long-term period (1979–2018).

- **The subject is interesting and useful and the paper is clearly written. However, the analysis lacks from a discussion about the uncertainties, which is crucial when using a model. In particular, dew yield estimation is very sensitive to the choice of the heat transfer coefficient (Eq. 6). An estimation of the final uncertainties in the expected dew yields should be therefore given and cited in the abstract and conclusion before publication.**

Response: Concerning the uncertainties of the model, a new subsection 3.3 has been added to the revised manuscript. In this new section we ran the model with 8 different parameterizations for heat transfer coefficient at 4 grid points (Table 6 and Figure 10) in different dew zones (illustrated in Figure 1) for the year 2000. So that the model was run 8 times for each station, 32 runs in total. In particular, the choose of Richard's parameterization seems to be reasonable for this study as it gives the dew yield which is close to average and are neither much lower nor much higher than the majority of other parameterizations and the largest differences occur for the very low values of daily dew yield. The result of uncertainty analysis has been also added in the abstract and conclusion. However, in an ideal situation we would compare our model results to observations, however, unfortunately observational data of dew formation in Iran is not available. Therefore, the accuracy of the modelled dew yields in comparison to observational data cannot be performed for this study. However, Vuollekoski et al. (2015, Section 2.3 and Fig. 2) and Atashi et al. (2021) presented detailed comparisons between results from this dew model and observations in other locations, where experimental dew data was available. The results of these studies were also added to this section in brief.

**Comment on hess-2021-54**
**Anonymous Referee #2**

This manuscript "Delineation of Dew Formation Zones in Iran Using Long-Term Model Simulations and Cluster Analysis" by Atashi et al. dealt with dew potential in Iran during 1978-2018. this is an important topic in such an arid and semi-arid region. The results can be of great importance for utilization of dew as an alternate source of water. The manuscript is well written and has a good flow of information that is easy to follow. I, therefore, foresee it suitable for the scope of this esteemed journal and recommend publication in the Print-Form after minor revisions.

- **I fully agree with the Anonymous Referee #1 (comment posted on 25 Mar 2021) and I share the same recommendation presented in the comment 'Referee Comment on hess-2021-54', Al-Kuisi Mustafa (posted on 24 Mar 2021).**

Response: We replied and considered all comments posted by "Al-Kuisi Mustafa".

- **Figure 2 is not needed. It is enough to indicate in the methods the best number of clusters is enough.**

Response: Thank you for this suggestion, however, we believe that Figure 2 is necessary evidence to support our choice of cluster number. A common question that readers have is "why did you chose X number of clusters" and we believe Figure 2 answers this much better than numbers in the text could. Furthermore, the issue of how many clusters we used was raised by another reviewer and therefore we kept Figure 2 in the revised version of this manuscript.

- **Figure 6 can be replotted to by showing the percentiles.**

Response: In order to highlight the information about the percentiles, the sentence describing Table 4 in lines 306-307 has been revised and a reference to Table 4 added in the caption of Figure 6.

- **The trend analysis shown in Section "3.2.2. Long-term temporal variation in dew formation zones" requires more discussion and relating the long-term changes to the possible reasons of the climate change. In order to support the discussion, I recommend moving the figures presented in the supplementary material to be a part of the main text of the manuscript. The authors might have seen them complementary, but I foresee them presenting important information to be presented side-by-side in the results.**

Response: we already discussed the long-term changes in dew formation in Iran and also investigated the possible reasons for this variation by doing a trend analysis on the related parameters in dew formation. Figure S3-S8 are some examples for different dew zones in Iran to support the discussion in section 3.3.1. these figures have been added to the main part of the text from the supplementary material as Figures 7 and Figure 8 and descriptions were added accordingly to each climate zone.